# Dynamics of hot Bose-Einstein condensates: stochastic Ehrenfest relations for number and energy damping

Rob G. McDonald[1,2], Peter S. Barnett[1,2], Fradom Atayee[1,2] and Ashton S. Bradley[1,2,⋆]

**1** Department of Physics, University of Otago, Dunedin, New Zealand
**2** The Dodd-Walls Centre for Photonic and Quantum Technologies

⋆ ashton.bradley@otago.ac.nz

## Abstract

Describing partially-condensed Bose gases poses a long-standing theoretical challenge. We present exact stochastic Ehrenfest relations for the stochastic projected Gross-Pitaevskii equation, including both number and energy damping mechanisms, and all projector terms that arise from the energy cutoff separating system from reservoir. We test the theory by applying it to the center of mass fluctuations of a harmonically trapped prolate system, finding close agreement between c-field simulations and analytical results. The formalism lays the foundation to analytically explore experimentally accessible hot Bose-Einstein condensates.

 Check for updates

# 1 Introduction

A system of Bose atoms with temperature $T$ undergoes a dramatic change in behavior at the critical temperature for the formation of a Bose-Einstein condensate (BEC), $T_c$. Far below the BEC transition $T \ll T_c$, a nearly pure BEC forms, consisting of a highly occupied many-body quantum state; in this regime dilute gas BEC are renowned both for their high experimental control, and precise theoretical description [1]. At temperatures $T \gg T_c$ thermal energy dominates, the quantum statistics are unimportant, and a Boltzman description captures the physical properties of the atoms. When $T \sim T_c$ the quantum statistics of the atoms are decisive, despite appreciable thermal energy. In a hot BEC $T \lesssim T_c$, competition between thermal and interaction effects leads to fragmentation of the condensate, and formation of vortices, solitons, and phononic excitations. A cooling quench across the transition can inject interesting excitations into the BEC that form as remnants of the broken $U(1)$ symmetry [2,3], and rich turbulent dynamics develop from the competition between thermal, quantum, and interaction effects, posing a challenge for theory.

At low temperatures that contain significant condensate and non-condensate fraction, mean field theory provides a useful description, upon which the GPE and its generalizations are based. The Zaremba-Nikuni-Griffin $U(1)$ symmetry breaking approach has proven well suited for practical calculations in this low-temperature regime [4], having the virtue that the interactions between condensate and thermal cloud, and their respective dynamics, are all included in the dynamical description. However, it's strength for low temperatures presents a limitation at high temperatures: despite notable successes, e.g. for collective modes [5], the symmetry-breaking ansatz has limited scope for describing strongly fluctuating systems containing large non-condensate fraction. Fortunately, the scope of the Gross-Pitaevskii equation (GPE) upon which ZNG is based goes much further than mean-field theory: GPE-like field equations appear naturally in phase-space representations of Bose gases [6], suggesting a generalization of quantum optical open systems theory [7] for describing hot Bose gases. Indeed, various generalized Gross-Pitaevskii theories have been developed for the high temperature regime, describing many modes that are weakly occupied, interacting, and partially coherent [8,9]. Devoid of the symmetry-breaking ansatz, these approaches have advantages for high-temperature work.

One approach capable of describing experiments across the phase transition is known as the

*stochastic projected Gross-Pitaevskii equation* (SPGPE) [10–14]. The SPGPE was developed as a synthesis of quantum kinetic theory [15] and the projected Gross-Pitaevskii equation [16], and provides a tractable approach for numerical simulations of hot matter-wave dynamics that includes all significant reservoir interaction processes. The SPGPE describes the evolution of a high-temperature partially condensed system within a classical field approximation, is valid on either side of the critical point, and has been used to quantitatively model the BEC phase transition [3], and BEC dynamics observed in high-temperature experiments [17]. The complete SPGPE includes a number-damping reservoir interaction (of Ginzburg-Landau type) described in previous works [3,17], and an additional interaction involving exchange of energy with the reservoir [2]; the latter is a number-conserving interaction that can in principle have a significant influence on dissipative evolution [9, 13, 18, 19], yet due to technical challenges, its physical effects have thus far been largely unexplored.

Ehrenfest's theorem [20] relates, for example, the time derivative of the expectation (with respect to the wavefunction) of momentum and position of a particle with the potential experienced by the particle. The Ehrenfest relations can be extended to the Gross-Pitaevskii fluid — essentially without modification — due to the net cancellation of two-body interaction forces. Previously, development of the number-damping SPGPE theory was aided by Ehrenfest relations [21]. Such relations for ensemble averages provide an essential validity check for numerical work, and physical consistency tests for the reservoir theory, including ensemble averages expected in equilibrium.

In this work we derive stochastic collective equations for the SPGPE including the effects of both number-damping and energy-damping. These stochastic Ehrenfest relations (SERs) form the extension of Ehrenfest relations to finite-temperature stochastic field theory of Bose-Einstein condensates. They extend beyond the scope of previously derived relations of this type [21] by explicitly retaining all noises and cutoff terms. Importantly, for many one-body operators the multiplicative noise in the energy-damped SPGPE is transformed into additive noise in the collective equations. This simplification opens the way for analytical treatments of a broad class of low-energy excitations in BEC including solitons, vortices, and collective modes, for both number- and energy-damping decay channels. As a test we apply the Ehrenfest relations to the center of mass fluctuations of a harmonically trapped system tightly confined along two spatial dimensions. We find that the analtyic solution of the SER for the center of mass is in close agreement with SPGPE simulations. The SPGPE theory of spinor and multi-component systems [14] enables these techniques to be applied to systems where dissipation can *only* proceed via energy damping.

The paper is structured as follows. The GPE, PGPE, and SPGPE are introduced in Sec. 2, and the Ehrenfest relations for the GPE and PGPE briefly reviewed. In Sec. 3 we apply Ito's formula for change of variables to derive SERs for the SPGPE. We then derive a set of simple fluctuation-dissipation relations describing equilibrium ensembles. In Sec. 4 we test the formalism on the center of mass motion of a harmonically trapped quasi-1D system. In Sec. 5 we discuss links between our work and relevant linterature, and give our conclusions.

## 2 Background

### 2.1 Gross-Pitaevskii equation

The Gross-Pitaevskii equation is the equation of motion for a scalar complex field evolving according to the Gross-Pitaevskii Hamiltonian

$$H = \int d^3\mathbf{r}\, \psi^*(\mathbf{r}, t) \left( -\frac{\hbar^2 \nabla^2}{2m} + V(\mathbf{r}, t) + \frac{g}{2} |\psi(\mathbf{r}, t)|^2 \right) \psi(\mathbf{r}, t), \tag{1}$$

where $V(\mathbf{r}, t)$ is an external potential and the interaction strength $g = 4\pi\hbar^2 a_s/m$ is the two-body interaction strength in the cold-collision regime [22] via the $s$-wave scattering length $a_s$ and atomic mass $m$. The Gross-Pitaevskii equation may be generated by taking the functional derivative of the Gross-Pitaevskii Hamiltonian

$$i\hbar\frac{\partial \psi(\mathbf{r}, t)}{\partial t} = \frac{\delta H}{\delta \psi^*(\mathbf{r}, t)} = L\psi(\mathbf{r}, t), \tag{2}$$

where the Gross-Pitaevskii operator is

$$L\psi(\mathbf{r}, t) \equiv \left(-\frac{\hbar^2\nabla^2}{2m} + V(\mathbf{r}, t) + g|\psi(\mathbf{r}, t)|^2\right)\psi(\mathbf{r}, t). \tag{3}$$

The total particle number

$$N = \int d^3\mathbf{r}\, \psi^*(\mathbf{r}, t)\psi(\mathbf{r}, t), \tag{4}$$

is conserved under the Hamiltonian evolution. Observables $A(t)$ of the system that are given by the expectation value of an operator $\hat{A}$ are defined by

$$A(t) \equiv \langle\hat{A}\rangle = \int d^3\mathbf{r}\, \langle\psi|\hat{A}|\mathbf{r}\rangle\psi(\mathbf{r}, t) = \int d^3\mathbf{r}\, \psi^*(\mathbf{r}, t)\langle\mathbf{r}|\hat{A}|\psi\rangle, \tag{5}$$

for example $\mathbf{R}(t) = \int d^3\mathbf{r}\, \psi^*(\mathbf{r})\mathbf{r}\psi(\mathbf{r})$ is the mean position of the atoms. The internal cancellation of s-wave interaction forces renders the Ehrenfest relations for the GPE identical to those of the Schrodinger equation [23]:

$$\frac{d\mathbf{R}(t)}{dt} = \frac{1}{m}\mathbf{P}(t), \tag{6}$$

$$\frac{d\mathbf{P}(t)}{dt} = -\langle\nabla V(\mathbf{r}, t)\rangle, \tag{7}$$

$$\frac{d\mathbf{L}(t)}{dt} = -\langle\mathbf{r}\times\nabla V(\mathbf{r}, t)\rangle, \tag{8}$$

$$\frac{dH(t)}{dt} = \left\langle\frac{\partial V(\mathbf{r}, t)}{\partial t}\right\rangle, \tag{9}$$

$$\frac{dN(t)}{dt} = 0, \tag{10}$$

for the position $\mathbf{R}(t)$, momentum $\mathbf{P}(t)$, angular momentum $\mathbf{L}(t)$, energy $H(t)$, and number $N(t)$ respectively. The focus of the present article is to develop a detailed theory of modifications to these collective equations that occur due to reservoir interactions in a Bose-Einstein condensate.

As is often convenient when changing representations, here we are using $|\psi\rangle$ to represent the abstract state ket associated with the pure-state wavefunction $\psi(\mathbf{r}) = \langle\mathbf{r}|\psi\rangle$. In the context of phase-space methods introduced below, it should be noted that we do not use this notation for the many-body quantum state, but just as a convenient shorthand for the wavefunction appearing in individual trajectories. Many-body expectation values are calculated as ensemble averages over trajectories of the stochastic field theory [9]. When generalizing the above definitions [(6)—(10)] to the many-body theory as described below, the average $\langle\cdot\rangle$ refers only to the wavefunction expectation over position coordinates. The combination of wavefunction and ensemble average will be denoted by $\langle\!\langle\cdot\rangle\!\rangle$.

## 2.2 Stochastic projected Gross-Pitaevskii equation

The derivation of the complete SPGPE for scalar BEC appeared in [10]. A later generalization allows for the possibility of multiple components and spins [14]. The derivation involves finding a master equation for the coherent region density operator. A series of approximations valid at high temperartures [21] then gives the high temperature regime master equation, which is mapped to an equivalent equation of motion for the multimode Wigner distribution function $W[\psi, \psi^*]$ using quantum to classical operator correspondences [7, 24]. The result of this is a generalized Fokker-Planck equation (FPE) of motion for the Wigner function that includes third-order functional derivates. An equation of motion for a quasi-probability distribution can only be mapped to an SDE if it takes the form of a Fokker-Planck equation, that is, only contains functional derivatives up to second order and has a positive semi-definite diffusion matrix. Futher progress can be made by neglecting the third-order derivatives, an approximation known as the *truncated Wigner approximation*. Such an approximation is physically well justified at high temperatures where thermal and classical noise dominate over quantum noise [25]. To establish notation and formulate the problem, we now summarize the key results.

### 2.2.1 Projector

The SPGPE requires a clear formulation of a projection operator that formally and numerically projects the nonlinear GPE dynamics into a low-energy subspace. To define the projector, the external potential is split into a time-invariant part and a time-dependent part $V(\mathbf{r}, t) \equiv V(\mathbf{r}) + \delta V(\mathbf{r}, t)$. The time independent part defines the single-particle Hamiltonian

$$H_{\mathrm{sp}} \equiv -\frac{\hbar^2 \nabla^2}{2m} + V(\mathbf{r}). \tag{11}$$

The basis of representation is chosen to be the eigenstates of this Hamiltonian, satisfying $H_{\mathrm{sp}} \phi_n(\mathbf{r}) = \epsilon_n \phi_n(\mathbf{r})$, where $n$ denotes the set of quantum numbers required to completely describe the single-particle basis states, with corresponding energy eigenvalues $\epsilon_n$.

The projector can be written in operator form as

$$\hat{\mathcal{P}} = \sum_{n \in C} |n\rangle\langle n|, \tag{12}$$

where the coherent region $C$ is defined as the set of basis states beneath the cutoff: $C \equiv \{n : \epsilon_n \leq \epsilon_{\mathrm{c}}\}$. In position space, this becomes an integral operator projecting an arbitrary function $F(\mathbf{r})$ into $C$, with action on the function $F(\mathbf{r})$ written as

$$\mathcal{P}\{F(\mathbf{r})\} \equiv \sum_{n \in C} \phi_n(\mathbf{r}) \int d^3\mathbf{r}' \phi_n^*(\mathbf{r}') F(\mathbf{r}') = \int d^3\mathbf{r}' \delta(\mathbf{r}, \mathbf{r}') F(\mathbf{r}'), \tag{13}$$

where we have made use of the projected Dirac-delta distribution

$$\delta(\mathbf{r}, \mathbf{r}') \equiv \sum_{n \in C} \phi_n(\mathbf{r}) \phi_n^*(\mathbf{r}') = \langle\mathbf{r}|\hat{\mathcal{P}}|\mathbf{r}'\rangle. \tag{14}$$

The orthogonal projector $\hat{\mathcal{Q}} = 1 - \hat{\mathcal{P}}$, has the action on the function $F(\mathbf{r})$

$$\mathcal{Q}\{F(\mathbf{r})\} \equiv \sum_{n \in I} \phi_n(\mathbf{r}) \int d^3\mathbf{r}' \phi_n^*(\mathbf{r}') F(\mathbf{r}'), \tag{15}$$

where the incoherent region $I$ is defined as $I \equiv \{n : \epsilon_n > \epsilon_{\mathrm{c}}\}$. Since $\hat{\mathcal{P}}$ is Hermitian, for any two states $|F\rangle, |G\rangle$, we have $\langle F|\hat{\mathcal{P}}|G\rangle = \langle F|\hat{\mathcal{P}}G\rangle = \langle\hat{\mathcal{P}}F|G\rangle$.

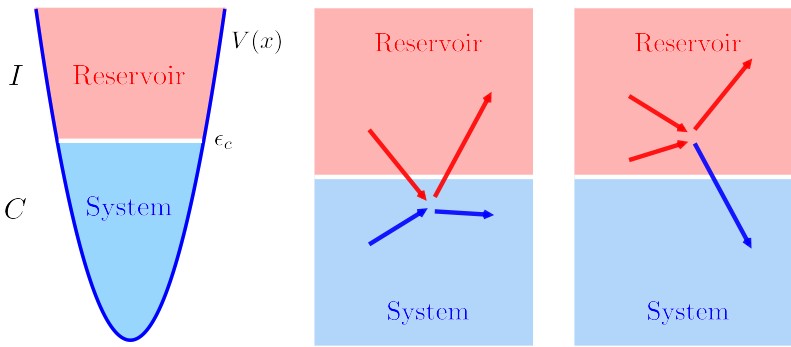

Figure 1: Schematic: separation of the hot Bose gas into a $C$-region system containing partially coherent modes with appreciable occupation, and $I$-region reservoir containing incoherent modes with small occupation (left); $C-I$ coupling via s-wave scattering: Energy-damping reservoir interaction (center); Number-damping reservoir interaction (right).

The classical field is represented as a sum over the basis states $\phi_n(\mathbf{r})$ with weight $\alpha_n(t)$

$$\psi(\mathbf{r}, t) = \sum_{n \in C} \alpha_n(t) \phi_n(\mathbf{r}), \tag{16}$$

where the field is now restricted to exist entirely within the coherent region. Without reservoir interactions, the projected Gross-Pitaevskii equation (PGPE) [16] is obtained by taking the *projected* functional derivative of $H$, where these are defined by

$$\frac{\bar{\delta}}{\bar{\delta}\psi(\mathbf{r}, t)} = \sum_{n \in C} \phi_n^*(\mathbf{r}) \frac{\partial}{\partial \alpha_n}, \qquad \frac{\bar{\delta}}{\bar{\delta}\psi^*(\mathbf{r}, t)} = \sum_{n \in C} \phi_n(\mathbf{r}) \frac{\partial}{\partial \alpha_n^*}. \tag{17}$$

The projected functional derivative is related to the regular functional derivative by

$$\frac{\bar{\delta}F[\psi, \psi^*]}{\bar{\delta}\psi(\mathbf{r}, t)} = \mathcal{P}^* \left\{ \frac{\delta F[\psi, \psi^*]}{\delta\psi(\mathbf{r}, t)} \right\}, \qquad \frac{\bar{\delta}F[\psi, \psi^*]}{\bar{\delta}\psi^*(\mathbf{r}, t)} = \mathcal{P} \left\{ \frac{\delta F[\psi, \psi^*]}{\delta\psi^*(\mathbf{r}, t)} \right\}. \tag{18}$$

In particular, the equation of motion for the c-field Eq. (16) is given by the PGPE:

$$i\hbar \frac{\partial \psi(\mathbf{r}, t)}{\partial t} = \frac{\bar{\delta}H}{\bar{\delta}\psi^*(\mathbf{r}, t)} = \mathcal{P}\{L\psi(\mathbf{r}, t)\}, \tag{19}$$

describing Hamiltonian evolution for a field that is formally restricted to the $C$ region. Defining the canonical momentum $\Pi(\mathbf{r}, t) \equiv i\hbar\psi(\mathbf{r}, t)^*$, and projected functional Poisson bracket of any two functionals $F[\psi, \Pi]$, $G[\psi, \Pi]$, as

$$\{F, G\} \equiv \int d^3\mathbf{r} \, \frac{\bar{\delta}F}{\bar{\delta}\psi(\mathbf{r})} \frac{\bar{\delta}G}{\bar{\delta}\Pi(\mathbf{r})} - \frac{\bar{\delta}F}{\bar{\delta}\Pi(\mathbf{r})} \frac{\bar{\delta}G}{\bar{\delta}\psi(\mathbf{r})}, \tag{20}$$

the PGPE follows the expected Hamiltonian structure

$$\partial_t \psi(\mathbf{r}) = \{\psi(\mathbf{r}), H\}, \qquad \partial_t \Pi(\mathbf{r}) = \{\Pi(\mathbf{r}), H\}. \tag{21}$$

Provided $\delta V \equiv 0$, $N$ and $H$ are both formally conserved by the PGPE: $\partial_t N = \{N, H\} = 0$, $\partial_t H = \{H, H\} = 0$. A short account of the former is instructive. Evaluating the projected derivatives, we have

$$\{N, H\} = \frac{1}{i\hbar} \int d^3\mathbf{r} \, \psi^*(\mathbf{r})(\mathcal{P}L\psi)(\mathbf{r}) - \psi(\mathbf{r})(\mathcal{P}L\psi)^*(\mathbf{r}). \tag{22}$$

Since $\mathcal{P}$ is hermitian, and $\mathcal{P}\psi \equiv \psi$ for projected fields, we then have $\{N, H\} \equiv 0$. As usual, additional conserved quantities may exist when the confining potential possesses additional symmetries, and numerical conservation requires that the basis also respect the symmetry. In practice this requires that the basis modes are eigenstates of the the symmetry generator.

### 2.2.2 Equations of motion

The PGPE gives Hamiltonian evolution of the coherent region field in isolation. In SPGPE c-field theory, the incoherent region is usually assumed to be thermalized and is thus treated as a reservoir that acts as a damping mechanism for the coherent region. Applying the truncated Wigner approximation to the Born-Markov master equation for the Bose field leads to the Fokker-Planck equation [10]

$$
\begin{aligned}
\frac{\partial W[\psi, \psi^*]}{\partial t} =& \int d^3\mathbf{r} \left[ -\frac{\bar{\delta}}{\bar{\delta}\psi(\mathbf{r})} \left( -\frac{i}{\hbar}(1 - i\gamma)(L - \mu)\psi(\mathbf{r}) - \frac{i}{\hbar} V_\varepsilon(\mathbf{r})\psi(\mathbf{r}) \right) + \text{h.c.} \right] W[\psi, \psi^*] \\
&+ \int d^3\mathbf{r} \left[ \frac{\gamma k_B T}{\hbar} \frac{\bar{\delta}^{(2)}}{\bar{\delta}\psi(\mathbf{r})\bar{\delta}\psi^*(\mathbf{r})} + \text{h.c.} \right] W[\psi, \psi^*] \\
&+ \int d^3\mathbf{r} \int d^3\mathbf{r}' \frac{k_B T}{\hbar} \varepsilon(\mathbf{r} - \mathbf{r}') \left[ \frac{\bar{\delta}}{\bar{\delta}\psi(\mathbf{r})} \psi(\mathbf{r}) \frac{\bar{\delta}}{\bar{\delta}\psi^*(\mathbf{r}')} \psi^*(\mathbf{r}') + \text{h.c.} \right] W[\psi, \psi^*] \\
&- \int d^3\mathbf{r} \int d^3\mathbf{r}' \frac{k_B T}{\hbar} \varepsilon(\mathbf{r} - \mathbf{r}') \left[ \frac{\bar{\delta}}{\bar{\delta}\psi(\mathbf{r})} \psi(\mathbf{r}) \frac{\bar{\delta}}{\bar{\delta}\psi(\mathbf{r}')} \psi(\mathbf{r}') + \text{h.c.} \right] W[\psi, \psi^*].
\end{aligned} \tag{23}
$$

The number-damping interction is parametrized by the number-damping rate $\gamma$ describing two-body collisions that transfer particles between $C$ and $I$. For a thermal Bose reservoir the dimensionless rate $\gamma$ is [11]

$$
\gamma = \frac{8a_s^2}{\lambda_{\text{dB}}^2} \sum_{j=1}^\infty \frac{e^{\beta\mu(j+1)}}{e^{2\beta\epsilon_c j}} \Phi\left[ \frac{e^{\beta\mu}}{e^{2\beta\epsilon_c}}, 1, j \right]^2, \tag{24}
$$

where $\lambda_{\text{dB}} = \sqrt{2\pi\hbar^2/mk_B T}$ is the thermal de Broglie wavelength, $\beta = 1/k_B T$, $\Phi[z, x, a] = \sum_{k=0}^\infty z^k/(a + k)^x$ is the *Lerch transcendent*.[1] Energy damping is described by the potential

$$
V_\varepsilon(\mathbf{r}) \equiv -\hbar \int d^3\mathbf{r}' \varepsilon(\mathbf{r} - \mathbf{r}') \nabla' \cdot \mathbf{j}(\mathbf{r}'), \tag{25}
$$

where $\mathbf{j}(\mathbf{r}) = i\hbar[\psi(\mathbf{r})\nabla\psi^*(\mathbf{r}) - \psi^*(\mathbf{r})\nabla\psi(\mathbf{r})]/2m$ is the current, and the *epsilon function*

$$
\varepsilon(\mathbf{r}) \equiv \frac{\mathcal{M}}{(2\pi)^3} \int d^3\mathbf{k}\, S(k) e^{i\mathbf{k}\cdot\mathbf{r}}, \tag{26}
$$

where $S(k) \equiv |\mathbf{k}|^{-1}$ is the scattering kernel. For identical bosons the rate constant $\mathcal{M}$ is given by [10, 13]

$$
\mathcal{M} = \frac{16\pi a_s^2 k_B T}{\hbar} \frac{1}{e^{\beta(\epsilon_c - \mu)} - 1}, \tag{27}
$$

while for distinct reservoir species, it is reduced by $1/2$ [14]. The scattering potential Eq. (25) describes the reservoir interaction involving transfer of energy and momentum without transfer of particles.

---

[1] A consequence of the high-energy cutoff ($\epsilon_c \sim 3\mu$) used in the SPGPE is that $\gamma$ is approximately independent of position, simplifying the SPGPE. Typical experimentally relevant values are of order $10^{-5} - 10^{-4}$ [11].

Direct solution of the Fokker-Planck equation of motion (23) is impractical numerically. Working with this FPE involving only derivatives up to second order, the Wigner function evolution equation can be mapped to an equivalent SDE for $\psi(\mathbf{r})$ [26], provided the diffusion matrix is positive semi-definite. For the FPE Eq. (23) this condition is always satisfied, and the stochastic projected Gross-Pitaevskii equation (SPGPE) is [10, 11, 14]

$$(\mathbf{S})d\psi(\mathbf{r}) = \mathcal{P}\left\{-\frac{i}{\hbar}(1-i\gamma)(L-\mu)\psi(\mathbf{r})dt - \frac{i}{\hbar}V_\varepsilon(\mathbf{r})\psi(\mathbf{r})dt + dW(\mathbf{r},t) + i\psi(\mathbf{r})dU(\mathbf{r},t)\right\}, \quad (28)$$

where the (complex) number-damping noise has the non-zero correlations

$$\langle dW(\mathbf{r},t)dW^*(\mathbf{r}',t)\rangle = \frac{2\gamma k_B T}{\hbar}\delta(\mathbf{r},\mathbf{r}')dt, \quad (29)$$

and the (real) energy-damping noise has non-zero correlations

$$\langle dU(\mathbf{r},t)dU(\mathbf{r}',t)\rangle = \frac{2k_B T}{\hbar}\varepsilon(\mathbf{r}-\mathbf{r}')dt. \quad (30)$$

The notation $(\mathbf{S})$ indicates that the SDE is to be interpreted as a stochastic integral in *Stratonovich* form [26], and thus at a given time $t$ the noises are not independent of the fields.

### 2.2.3 Energy cutoff and the single-particle basis

Before presenting our results, we remark that there exists a variety of interpretations in the literature of a central feature of *c*-field theory: what constitutes a cutoff, and how should it be imposed in practice? In some cases any convenient numerical basis of representation is truncated. However, truncation in an arbitrary basis will not usually generate a consistent energy cutoff, and some care should be taken when choosing a basis of representation. In particular, if a highly inappropriate basis of representation is used, then thermal noise driving the *c*-field modes will generate thermalization artifacts [21]. In our approach to the SPGPE we impose the cutoff in the single-particle basis generated by the time-independent confining potential. There is a clear physical motivation for this choice (clearly articulated by the pioneers of the PGPE): at sufficiently high energy — usually of order 2-3$\mu$ — the interacting many-body problem is approximately diagonal in the single-particle basis. Thus a well-chosen cutoff in the single-particle basis is also an approximate energy cutoff for the many-body system, precisely what is required for its separation into coherent and incoherent subspaces. These considerations will feature below in our discussion of equilibrium properties of the SPGPE.

## 3 Stochastic Ehrenfest relations

Ehrenfest relations for the stochastic field can be derived by casting the SPGPE in *Ito* form. The noises and fields are then uncorrelated, allowing change of variables using the rules of Ito calculus.

### 3.1 Ito form of the SPGPE

Recasting the SPGPE in Ito form involves reordering the projected function derivatives in the final term of the FPE and mapping the equation to a new SDE.

The FPE is equivalent to

$$
\begin{aligned}
\frac{\partial W[\psi,\psi^*]}{\partial t} =& \int d^3\mathbf{r}\left[-\frac{\bar{\delta}}{\bar{\delta}\psi(\mathbf{r})}\left(-\frac{i}{\hbar}(1-i\gamma)(L-\mu)\psi(\mathbf{r})-\frac{i}{\hbar}V_\varepsilon(\mathbf{r})\psi(\mathbf{r})\right)+\text{h.c.}\right]W[\psi,\psi^*]\\
&+\int d^3\mathbf{r}\left[-\frac{\bar{\delta}}{\bar{\delta}\psi(\mathbf{r})}\left(-\frac{k_B T}{\hbar}\int d^3\mathbf{r}'\varepsilon(\mathbf{r}-\mathbf{r}')\delta(\mathbf{r},\mathbf{r}')\psi(\mathbf{r}')\right)+\text{h.c.}\right]W[\psi,\psi^*]\\
&+\int d^3\mathbf{r}\left[\frac{\gamma k_B T}{\hbar}\frac{\bar{\delta}^{(2)}}{\bar{\delta}\psi(\mathbf{r})\bar{\delta}\psi^*(\mathbf{r})}+\text{h.c.}\right]W[\psi,\psi^*]\\
&+\int d^3\mathbf{r}\int d^3\mathbf{r}'\frac{k_B T}{\hbar}\varepsilon(\mathbf{r}-\mathbf{r}')\left[\frac{\bar{\delta}^{(2)}}{\bar{\delta}\psi(\mathbf{r})\bar{\delta}\psi^*(\mathbf{r}')}\psi(\mathbf{r})\psi^*(\mathbf{r}')+\text{h.c.}\right]W[\psi,\psi^*]\\
&-\int d^3\mathbf{r}\int d^3\mathbf{r}'\frac{k_B T}{\hbar}\varepsilon(\mathbf{r}-\mathbf{r}')\left[\frac{\bar{\delta}^{(2)}}{\bar{\delta}\psi(\mathbf{r})\bar{\delta}\psi(\mathbf{r}')}\psi(\mathbf{r})\psi(\mathbf{r}')+\text{h.c.}\right]W[\psi,\psi^*],\quad(31)
\end{aligned}
$$

which maps to the Ito SPGPE

$$
\begin{aligned}
(\mathbf{I})d\psi(\mathbf{r})=\mathcal{P}\Big\{&-\frac{i}{\hbar}(1-i\gamma)(L-\mu)\psi(\mathbf{r})dt-\frac{i}{\hbar}V_\varepsilon(\mathbf{r})\psi(\mathbf{r})dt\\
&-\frac{k_B T}{\hbar}\int d^3\mathbf{r}'\varepsilon(\mathbf{r}-\mathbf{r}')\delta(\mathbf{r},\mathbf{r}')\psi(\mathbf{r}')dt+dW(\mathbf{r},t)+i\psi(\mathbf{r})dU(\mathbf{r},t)\Big\},\quad(32)
\end{aligned}
$$

where the first term in the second line is called the *Stratonovich correction*. In contrast to the Stratonovich SPGPE, in the Ito SPGPE [denoted (**I**)] the noises are always independent of the fields at a given time $t$. This formulation has distinct advantages for formal manipulation that we exploit below.

## 3.2 Functional Change of Variables

Consider any functional of the projected fields $A\equiv A[\psi,\psi^*,t]$. Using the rules of Ito calculus, we can find the an SDE for $A$ in the form

$$
\begin{aligned}
(\mathbf{I})dA[\psi,\psi^*,t]=&\frac{\partial A[\psi,\psi^*,t]}{\partial t}dt+\int d^3\mathbf{r}\left[\frac{\bar{\delta}A[\psi,\psi^*,t]}{\bar{\delta}\psi(\mathbf{r})}d\psi(\mathbf{r})+\text{h.c.}\right]\\
&+\int d^3\mathbf{r}\int d^3\mathbf{r}'\left[\frac{\bar{\delta}^{(2)}A[\psi,\psi^*,t]}{\bar{\delta}\psi^*(\mathbf{r}')\bar{\delta}\psi(\mathbf{r})}\delta(\mathbf{r},\mathbf{r}')+\text{h.c.}\right]\frac{\gamma k_B T}{\hbar}dt\\
&+\int d^3\mathbf{r}\int d^3\mathbf{r}'\left[\frac{\bar{\delta}^{(2)}A[\psi,\psi^*,t]}{\bar{\delta}\psi^*(\mathbf{r}')\bar{\delta}\psi(\mathbf{r})}\psi^*(\mathbf{r}')\psi(\mathbf{r})+\text{h.c.}\right]\varepsilon(\mathbf{r}-\mathbf{r}')\frac{k_B T}{\hbar}dt\\
&-\int d^3\mathbf{r}\int d^3\mathbf{r}'\left[\frac{\bar{\delta}^{(2)}A[\psi,\psi^*,t]}{\bar{\delta}\psi(\mathbf{r}')\bar{\delta}\psi(\mathbf{r})}\psi(\mathbf{r}')\psi(\mathbf{r})+\text{h.c.}\right]\varepsilon(\mathbf{r}-\mathbf{r}')\frac{k_B T}{\hbar}dt,\quad(33)
\end{aligned}
$$

where we consistently include all terms up to order $dt$, including terms quadratic in the noises that generate second derivatives. We require the property of the epsilon function that it is diagonal in $k$-space:

$$
\int d^3\mathbf{r}\int d^3\mathbf{u}\,F(\mathbf{r})G(\mathbf{u})\varepsilon(\mathbf{r}-\mathbf{u})=\mathcal{M}\int d^3\mathbf{k}\,S(k)\mathcal{F}[F^*(\mathbf{r})]^*\mathcal{F}[G(\mathbf{r})],\quad(34)
$$

where the Fourier transform is defined as

$$
\mathcal{F}[f(\mathbf{r})]\equiv\frac{1}{(2\pi)^{3/2}}\int d^3\mathbf{r}\,f(\mathbf{r})e^{-i\mathbf{k}\cdot\mathbf{r}},\quad(35)
$$

a simplification that we will make use of below. Using

$$\frac{\bar{\delta}A[\psi,\psi^*,t]}{\bar{\delta}\psi(\mathbf{r})} = \frac{\delta A[\psi,\psi^*,t]}{\delta\psi(\mathbf{r})} - \mathcal{Q}^*\left(\frac{\delta A[\psi,\psi^*,t]}{\delta\psi(\mathbf{r})}\right), \tag{36}$$

we rewrite (33) as a stochastic Ehrenfest equation with additional terms due to the projector

$$
\begin{aligned}
(\mathbf{I})dA[\psi,\psi^*,t] = {}&\frac{\partial A[\psi,\psi^*,t]}{\partial t}dt + \frac{2}{\hbar}\mathrm{Im}\int d^3\mathbf{r}\frac{\delta A[\psi,\psi^*,t]}{\delta\psi(\mathbf{r})}(L-\mu)\psi(\mathbf{r})dt + Q_A^H \\
&- \frac{2\gamma}{\hbar}\mathrm{Re}\int d^3\mathbf{r}\frac{\delta A[\psi,\psi^*,t]}{\delta\psi(\mathbf{r})}(L-\mu)\psi(\mathbf{r})dt + Q_A^\gamma \\
&- 2\mathcal{M}\,\mathrm{Im}\int d^3\mathbf{k}\,S(k)\mathcal{F}\left[\frac{\delta A[\psi,\psi^*,t]}{\delta\psi^*(\mathbf{r})}\psi^*(\mathbf{r})\right]^*\mathcal{F}[\nabla\cdot\mathbf{j}(\mathbf{r})]dt + Q_A^\varepsilon \\
&+ \frac{2\gamma k_\mathrm{B}T}{\hbar}\mathrm{Re}\int d^3\mathbf{r}\int d^3\mathbf{u}\,\delta(\mathbf{u},\mathbf{r})\frac{\delta^{(2)}A[\psi,\psi^*,t]}{\delta\psi^*(\mathbf{r})\delta\psi(\mathbf{u})}dt + dA^\varepsilon \\
&+ dW_\gamma^A(t) + dW_\varepsilon^A(t), 
\end{aligned}
\tag{37}
$$

where the noise correlations are

$$\langle dW_\gamma^A(t)dW_\gamma^A(t)\rangle = \left(\frac{4\gamma k_\mathrm{B}T}{\hbar}\int d^3\mathbf{r}\left|\frac{\delta A[\psi,\psi^*,t]}{\delta\psi(\mathbf{r})}\right|^2 + D_A^\gamma\right)dt, \tag{38}$$

$$\langle dW_\varepsilon^A(t)dW_\varepsilon^A(t)\rangle = \left(\frac{8\mathcal{M}k_\mathrm{B}T}{\hbar}\int d^3\mathbf{k}\,k^{-1}\left|\mathcal{F}\left[\mathrm{Im}\left\{\frac{\delta A[\psi,\psi^*,t]}{\delta\psi(\mathbf{r})}\psi(\mathbf{r})\right\}\right]\right|^2 + D_A^\varepsilon\right)dt, \tag{39}$$

and the Hamiltonian, number-damping, and energy-damping drift projector terms are given in Appendix A.1. The number-damping and energy-damping each have a corresponding drift, diffusion, and trace[2] term. It is worth remarking that that the general expression obtained above allows for any observable. Furthermore, it is well-suited for significant simplification via an ansatz for the wavefunction, provided the integrals may be evaluated. In the remainder of this paper we explore the consequences of this formulation, providing explicit stochastic equations for particular observables of interest.

## 3.3 One-body operators

If we restrict our attention to one-body operators, the projected functional calculus assumes a particularly simple form. Consider a quantity that is the pure-state trace of a one-body operator

$$A[\psi,\psi^*] = \langle\psi|\hat{A}|\psi\rangle = \int d^3\mathbf{r}\langle\psi|\hat{A}|\mathbf{r}\rangle\psi(\mathbf{r}), \tag{40}$$

where the operator $\hat{A}$ is independent of $\psi,\psi^*$. We emphasize that this quantity is spatial average for the pure state wavefunction associated with each trajectory, and does not represent the many-body expectation value found by tracing over the density matrix (the ensemble average). The non-zero projected functional derivatives are

$$\frac{\bar{\delta}A[\psi,\psi^*,t]}{\bar{\delta}\psi(\mathbf{r})} = \langle\psi|\hat{\mathcal{P}}\hat{A}\hat{\mathcal{P}}|\mathbf{r}\rangle = \frac{\delta A_\mathcal{P}[\psi,\psi^*,t]}{\delta\psi(\mathbf{r})}, \tag{41}$$

$$\frac{\bar{\delta}^{(2)}A[\psi,\psi^*,t]}{\bar{\delta}\psi^*(\mathbf{r}')\bar{\delta}\psi(\mathbf{r})} = \langle\mathbf{r}'|\hat{\mathcal{P}}\hat{A}\hat{\mathcal{P}}|\mathbf{r}\rangle = \frac{\delta^{(2)}A_\mathcal{P}[\psi,\psi^*,t]}{\delta\psi^*(\mathbf{r}')\delta\psi(\mathbf{r})}. \tag{42}$$

---

[2]We refer to $dA^\varepsilon$ loosely as the energy-damping trace term, despite the fact it is not strictly a trace; it is the energy-damping counterpart to the number-damping trace term, which is a true trace.

The projected functional derivatives corresponding to the operator $\hat{A}$ are equivalent to the regular functional derivatives of the totally projected operator $\hat{A}_{\mathcal{P}} \equiv \hat{\mathcal{P}}\hat{A}\hat{\mathcal{P}}$; the change of variables is then simple to construct. Again writing in the form of a wave function average with additional projector terms, the SDE of the observable $A$ is

$$
\begin{aligned}
(\mathbf{I})dA(t) = & \left\langle \frac{\partial \hat{A}}{\partial t} \right\rangle dt + \frac{1}{i\hbar} \left\langle [\hat{A}, \hat{H}_{\mathrm{sp}}] \right\rangle dt + \frac{1}{\hbar} 2\mathrm{Im} \int d^3\mathbf{r} \langle \psi|\hat{A}|\mathbf{r}\rangle (gn(\mathbf{r}) + \delta V(\mathbf{r}, t) - \mu)\psi(\mathbf{r}) dt \\
& - \frac{\gamma}{\hbar} 2\mathrm{Re} \int d^3\mathbf{r} \langle \psi|\hat{A}|\mathbf{r}\rangle (L - \mu)\psi(\mathbf{r}) dt + \frac{2\gamma k_{\mathrm{B}} T}{\hbar} \mathrm{Tr}\left(\hat{A}\hat{\mathcal{P}}\right) dt \\
& - 2\mathcal{M} \int d^3\mathbf{k}\, S(k) \mathcal{F}\left[\mathrm{Im}\langle \psi|\hat{A}|\mathbf{r}\rangle \psi(\mathbf{r})\right]^* \mathcal{F}[\nabla \cdot \mathbf{j}(\mathbf{r})] dt \\
& + dW_\gamma^A(t) + dW_\varepsilon^A(t) \\
& + dA^\varepsilon + Q_A^H + Q_A^\gamma + Q_A^\varepsilon,
\end{aligned}
\tag{43}
$$

where the noise correlations are

$$
\langle dW_\gamma^A(t) dW_\gamma^A(t) \rangle = \left( \frac{4\gamma k_{\mathrm{B}} T}{\hbar} \langle \hat{A}^2 \rangle + D_A^\gamma \right) dt,
\tag{44}
$$

$$
\langle dW_\varepsilon^A(t) dW_\varepsilon^A(t) \rangle = \left( \frac{8\mathcal{M} k_{\mathrm{B}} T}{\hbar} \int d^3\mathbf{k}\, k^{-1} \left| \mathcal{F}\left[ \mathrm{Im}\left\{ \langle \psi|\hat{A}|\mathbf{r}\rangle \psi(\mathbf{r}) \right\} \right] \right|^2 + D_A^\varepsilon \right) dt,
\tag{45}
$$

and the projector terms $dA^\varepsilon - Q_A^\varepsilon$ are given in Appendix A. We have expressed Eq. (43) in terms of the commutator $[\hat{A}, \hat{H}_{\mathrm{sp}}]$ to show the connection to the standard Ehrenfest relations.

## 3.4 Noise correlations for multiple moments

When considering more than one moment, one must take care when considering the noise terms that arise. Let $B = B[\psi, \psi^*, t]$ be another moment of the SPGPE with evolution described by Eq. (37). The correlations between the noises corresponding to $A$ and the noises corresponding to $B$ are

$$
\langle dW_\gamma^A(t) dW_\gamma^B(t) \rangle = \frac{4\gamma k_{\mathrm{B}} T}{\hbar} \mathrm{Re} \int d^3\mathbf{r} \int d^3\mathbf{r}' \frac{\bar{\delta} A[\psi, \psi^*, t]}{\bar{\delta}\psi(\mathbf{r})} \frac{\bar{\delta} B[\psi, \psi^*, t]}{\bar{\delta}\psi^*(\mathbf{r}')} \delta(\mathbf{r}, \mathbf{r}') dt
\tag{46}
$$

$$
\begin{aligned}
\langle dW_\varepsilon^A(t) dW_\varepsilon^B(t) \rangle = & \frac{8\mathcal{M} k_{\mathrm{B}} T}{\hbar} \int d^3\mathbf{k}\, k^{-1} \mathcal{F}\left[ \mathrm{Im}\left\{ \frac{\bar{\delta} A[\psi, \psi^*, t]}{\bar{\delta}\psi(\mathbf{r})} \psi(\mathbf{r}, t) \right\} \right]^* \\
& \times \mathcal{F}\left[ \mathrm{Im}\left\{ \frac{\bar{\delta} B[\psi, \psi^*, t]}{\bar{\delta}\psi(\mathbf{r})} \psi(\mathbf{r}, t) \right\} \right] dt,
\end{aligned}
\tag{47}
$$

for number-damping and energy-damping respectively.

The special case of one-body operators allows significant simplification: for moments that are the expectation of a one-body operator the correlations reduce to

$$
\langle dW_\gamma^A(t) dW_\gamma^B(t) \rangle = \frac{4\gamma k_{\mathrm{B}} T}{\hbar} \mathrm{Re}\langle \hat{A}\hat{\mathcal{P}}\hat{B} \rangle dt,
\tag{48}
$$

$$
\begin{aligned}
\langle dW_\varepsilon^A(t) dW_\varepsilon^B(t) \rangle = & \frac{8\mathcal{M} k_{\mathrm{B}} T}{\hbar} \int d^3\mathbf{k}\, k^{-1} \mathcal{F}\left[ \mathrm{Im}\left\{ \langle \psi|\hat{A}\hat{\mathcal{P}}|\mathbf{r}\rangle \psi(\mathbf{r}, t) \right\} \right]^* \\
& \times \mathcal{F}\left[ \mathrm{Im}\left\{ \langle \psi|\hat{B}\hat{\mathcal{P}}|\mathbf{r}\rangle \psi(\mathbf{r}, t) \right\} \right] dt,
\end{aligned}
\tag{49}
$$

for number-damping and energy-damping respectively.

### 3.5 Finite-temperature stochastic Ehrenfest relations

We consider the Ehrenfest relations for position, momentum, angular momentum, grand canonical energy, and coherent region particle number. The complete set of *stochastic Ehrenfest relations* for the SPGPE are

$$
\begin{aligned}
(\mathbf{I})dR_j(t) = {} & \frac{1}{m}P_j(t)dt - \frac{2\gamma}{\hbar}\mathrm{Re}\left\langle \hat{r}_j(L-\mu)\right\rangle dt + \frac{2\gamma k_\mathrm{B}T}{\hbar}\mathrm{Tr}\left(\hat{r}_j\hat{\mathcal{P}}\right)dt \\
& + dW_\gamma^{R_j}(t) + dW_\varepsilon^{R_j}(t) \\
& + Q_{r_j}^H + Q_{r_j}^\gamma + Q_{r_j}^\varepsilon + dR_j^\varepsilon,
\end{aligned} \tag{50a}
$$

$$
\begin{aligned}
(\mathbf{I})dP_j(t) = {} & -\left\langle \partial_j V(\mathbf{r},t)\right\rangle dt - \frac{2\gamma}{\hbar}\mathrm{Re}\left\langle \hat{p}_j(L-\mu)\right\rangle dt + \frac{2\gamma k_\mathrm{B}T}{\hbar}\mathrm{Tr}\left(\hat{p}_j\hat{\mathcal{P}}\right)dt \\
& - \hbar\mathcal{M}\int d^3\mathbf{k}\,S(k)\mathcal{F}\left[\partial_j n(\mathbf{r})\right]^*\mathcal{F}\left[\nabla\cdot\mathbf{j}(\mathbf{r})\right]dt \\
& + dW_\gamma^{P_j}(t) + dW_\varepsilon^{P_j}(t) \\
& + Q_{p_j}^H + Q_{p_j}^\gamma + Q_{p_j}^\varepsilon + dP_j^\varepsilon,
\end{aligned} \tag{50b}
$$

$$
\begin{aligned}
(\mathbf{I})dL_j(t) = {} & -\left\langle \left(r_{j+1}\partial_{-1} - r_{j-1}\partial_{j+1}\right)V(\mathbf{r},t)\right\rangle dt - \frac{2\gamma}{\hbar}\mathrm{Re}\left\langle \hat{l}_j(L-\mu)\right\rangle dt \\
& + \frac{2\gamma k_\mathrm{B}T}{\hbar}\mathrm{Tr}\left(\hat{l}_j\hat{\mathcal{P}}\right)dt \\
& - \hbar\mathcal{M}\int d^3\mathbf{k}\,S(k)\mathcal{F}\left[\left(r_{j+1}\partial_{-1} - r_{j-1}\partial_{j+1}\right)n(\mathbf{r})\right]^*\mathcal{F}\left[\nabla\cdot\mathbf{j}(\mathbf{r})\right]dt \\
& + dW_\gamma^{L_j}(t) + dW_\varepsilon^{L_j}(t) \\
& + Q_{l_j}^H + Q_{l_j}^\gamma + Q_{l_j}^\varepsilon + dL_j^\varepsilon,
\end{aligned} \tag{50c}
$$

$$
\begin{aligned}
(\mathbf{I})dK(t) = {} & \left\langle \frac{\partial V(\mathbf{r},t)}{\partial t}\right\rangle dt - \frac{2\gamma}{\hbar}\mathrm{Re}\left\langle (L-\mu)^2\right\rangle dt + \frac{2\gamma k_\mathrm{B}T}{\hbar}\mathrm{Tr}\left(\hat{\mathcal{P}}(L-\mu)\hat{\mathcal{P}}\right)dt \\
& - \hbar\mathcal{M}\int d^3\mathbf{k}\,S(k)\left|\mathcal{F}\left[\nabla\cdot\mathbf{j}(\mathbf{r})\right]\right|^2 dt \\
& + dW_\gamma^K(t) + dW_\varepsilon^K(t) \\
& + Q_K^H + Q_K^\gamma + Q_K^\varepsilon + dK^\varepsilon,
\end{aligned} \tag{50d}
$$

$$
(\mathbf{I})dN(t) = -\frac{2\gamma}{\hbar}\mathrm{Re}\langle (L-\mu)\rangle dt + \frac{2\gamma k_\mathrm{B}T\mathcal{N}}{\hbar}dt + dW_\gamma^N(t), \tag{50e}
$$

where the noise correlators are given by Eqs. (46),(47). This set of equations are our main result. They take the form of generalized Ehrenfest relations with additional damping and noise terms arising from the reservoir coupling processes. They provide a starting point for finding analytic descriptions of hot BEC dynamics, and also provide tests for numerical consistency of SPGPE simulations. We make the following remarks:

   i) *Ensemble averages.*—Neglecting energy-damping and taking the average over the noise (in Ito form the fields and noises are uncorrelated), we immediately recover the Ehrenfest relations for the number-damped SPGPE as found in [21]. Those Ehrenfest relations described the evolution of ensemble avarages, whereas in the present formulation we retain all noises in the collective equations.

   ii) *Multiplicative noise.*—Superficially, the multiplicative noise in the SPGPE Eq. (28) may appear to have been transformed into additive noise. However, we emphasize that mul-

tiplicative noises remain present in terms of the form $\sqrt{\langle \hat{A} \rangle} dW_j$, since $\langle \hat{A} \rangle$ is not a noise average.

iii) *Additive noise.*—Reduction to additive noise can be achieved in special cases where the system can be well-described by a suitable physically motivated ansatz wavefunction. If such an ansatz is available, reduction to an additive noise SDE provides a significant simplification that can enable analytical progress [19].

iv) *Projector terms.*—The projector terms are all consistently accounted for, and in general contribute additional noises. However, provided the basis of projection is properly chosen, their effect is typically only a small correction. Testing whether such terms are negligible provides a useful consistency test for a well-chosen cutoff.

v) *Thermal equilibrium.*—Formally it is known that trajectories of the SPGPE will approach a grand canonical ensemble [10] describing the thermal equilibrium properties of the system. The c-field ensemble is then drawn from the equilibrium Wigner functional

$$W[\psi, \psi^*] \propto \exp(-\beta(H - \mu N)). \tag{51}$$

In the SPGPE, the particular ensemble depends upon the ratio $\mathcal{M}/\gamma$, with canonical equilibrium reached in the limit $\mathcal{M} \gg \gamma$, and grand-canonical equilibrium reached in the opposite limit. The equilibrium ensemble satisfies a set of fluctuation-dissipation relations, as may be derived by applying the steady-state condition to Eq. (50). In each case there is a balance between thermal noise and the damping. For example, for pure number damping it is easily shown that $\langle\langle L - \mu \rangle\rangle = k_B T \mathcal{N}$ for $\mathcal{N} \equiv \mathrm{Tr}\,\hat{\mathcal{P}}$ modes in the $C$-region. The correct balance between damping and noise is manifest through the absence of the dissipation parameters from such expressions, consistent with their absence from the equilibrium ensemble Eq. (51). We note that ensemble (in)equivalence [27] and the choice of an optimal cutoff [28] are both active areas of research.

# 4 Application to center of mass fluctuations

As a test of the formalism we consider center of mass oscillations in a quasi-1D harmonically trapped system with frequency for transverse trapping $\omega_\perp$ such that $\omega_\perp \gg \omega$, the frequency for the axial trap. Our primary aim is to test the validity of our analytical approach against c-field simulations. The system we choose is particularly simple and thus amenable to analytic treatment of the SERs. However, we must take some care in interpreting the reservoir theory in this case, as the Bose gas in a harmonic trap obeys Kohn's theorem.

Kohn's theorem states that in a harmonically trapped system the center of mass undergoes bulk oscillations about the trap center at the trapping frequency. While the second-quantized field theory satisfies Kohn's theorem, the best available c-field reservoir theory describes the $I$ region as time independent, and hence violates Kohn's theorem. C-field theory is thus currently best suited to systems for which a *time-independent* high-energy thermal reservoir is a reasonable approximation. We emphasize that the low-energy c-field is a dynamical treatment of both condensate and non-condensate; above a high-energy cutoff the dynamics are typically not included due to technical limitations.

While the time-independent reservoir approximation (TIRA) is not strictly applicable for scalar BEC in the purely harmonic trap, there are a number of physical systems where it is applicable: Kohn's theorem does not apply to a scalar Bose gas held in a harmonic trap if the trap becomes non harmonic at high energy. The theorem is also inapplicable for a harmonically trapped system if the reservoir consists of second atomic species confined by a different

trapping potential, as may occur during sympathetic cooling. Furthermore, any system that is not harmonically trapped will not obey Kohn's theorem, and is thus potentially amenable to the TIRA. Examples systems in non-harmonic traps for which the theory is applicable include vortex decay in hard-wall confinement [46], soliton decay in a 1D toroidal trap (where the present approach was first used) [19], and persistent current formation in a 3D toroidal trap, where SPGPE simulations [17] compare well with experiment [44].

In this work our approach is simply to test the formalism on a simple model system within the TIRA by integrating out the spatial degrees of freedom to find effective stochastic equations of motion for the center of mass. We stress that the TIRA approached used here is physically valid for (at least) two scenarios if immediate interest: non-harmonic trapping at high energies for a scalar BEC, and sympathetic cooling involving two BEC components in different harmonic traps [14][3]. An effective one-dimensional theory can be found in the prolate trapping regime where the reservoir remains three dimensional[4], and the low-dimensional subspace is well-described by projecting onto the transverse ground state of the confining potential.

The one dimensional SPGPE assumes an identical functional form to the three dimensional SPGPE, but with dimensionally reduced damping and noise coefficients [18]. The number-damping term is only changed by the replacement $g \to g_{1D}$, as are the Hamiltonian terms. The scattering kernel in the 1D reduction of Eq. (26) becomes

$$S_1(k) = \frac{1}{\sqrt{8\pi a_\perp^2}} \mathrm{erfcx}\left(\frac{|k|a_\perp}{\sqrt{2}}\right), \tag{52}$$

where $\mathrm{erfcx}(q) \equiv e^{q^2}\mathrm{erfc}(q)$ is the scaled complementary error function, and $a_\perp \equiv \sqrt{\hbar/m\omega_\perp}$ is the transverse harmonic oscillator length, much smaller than $a_\omega = \sqrt{\hbar/m\omega}$. The system is assumed to be sufficiently condensed such that the center of mass motion can be approximated using a Thomas-Fermi wavefunction ansatz. We do not require that this is a good approximation — indeed, the high-temperature regime requires that it must not be. We merely require that it should approximate the center of mass motion. The Thomas-Fermi wave function allowing for arbitrary variations in the center of mass position $x(t)$ and momentum $p(t)$ is

$$\psi(x) = \sqrt{\frac{\mu}{g}}\sqrt{1 - \frac{(x - x(t))^2}{R^2}} \exp\left[\frac{ip(t)x}{\hbar}\right]. \tag{53}$$

Using this ansatz, the integrals for Eqs. (50) can be evaluated analytically.[5]

## 4.1 Stochastic equations for center of mass

In the interest of brevity, we outline the derivation here, providing details in Appendix B. We first use the Thomas-Fermi ansatz Eq. (53) and evaluate the spatial integrals in (50a), (50b) to derive stochastic equations for $x$ and $p$ that are exact within the Thomas-Fermi approximation. The projector terms, given in Appendix B.1, severely constrain *exact* analytic progress beyond this point. Provided these terms may be safely neglected, we can arrive at an approximate set of equations that can be solved analytically. For consistent c-field simulations, neglecting the projector terms in the SERs will always be a good approximation, since for a well-chosen cutoff the mode population near the cutoff will be relatively small. Since we are considering

---

[3]The theory introduced here can be developed into a first-principles treatment for a particular atom number and temperature by self-consistent calculation of the reservoir interaction parameters [3, 13, 17].

[4]Without this restriction the complexity of the reservoir interactions increases significantly and we do not pursue this regime further here.

[5]We note that in the basis of harmonic oscillator modes used to represent the $C$-region the trace terms in the Ehrenfest relations for position and momentum vanish.

equilibrium states, we can assume that the values of $x(t)$ and $p(t)$ will also be small relative to the characteristic harmonic oscillator length and momentum scales; we test our assumptions by solving the SPGPE numerically. It is convenient to rewrite the equations of motion as a single equation of motion for the dimensionless complex variable

$$z(t) = \sqrt{\frac{m\omega}{2\hbar}}x(t) + \frac{i}{\sqrt{2\hbar m\omega}}p(t), \tag{54}$$

and neglect terms of higher order than linear in $z(t)$, $z^*(t)$, to find the simpler stochastic equations given in Appendix B, Eq. (76). Assuming it is physically consistent to neglect projector terms in (76), (77a), we arrive at the equation of motion

$$(\mathbf{I})dz(t) = -i\omega z(t)dt - (\Lambda_\gamma + \Lambda_\varepsilon)z(t)dt - (\Lambda_\gamma - \Lambda_\varepsilon)z^*(t)dt + dW_\gamma^z(t) + dW_\varepsilon^z(t), \tag{55}$$

with noise correlations

$$\langle dW_\gamma^z(t)dW_\gamma^z(t)\rangle = \frac{\mathcal{D}_\gamma}{2\hbar m\omega^3}dt, \tag{56a}$$

$$\langle dW_\varepsilon^z(t)dW_\varepsilon^z(t)\rangle = -\frac{m\mathcal{D}_\varepsilon}{2\hbar\omega}dt. \tag{56b}$$

The equation of motion describes simple harmonic motion with two sources of damping and noise.

## 4.2 Analytic and numerical solutions

In this section we present an analytical treatment of the stochastic equations discussed in Section 4.1, and compare the results with numerical simulations of the 1D SPGPE. The analytic approach offers a simple intuitive picture of the dynamics of the center of mass, for both number and energy damping mechanisms, and for both dissipation and noise.

Neglecting the projector terms, we can write the coupled differential equation as a vector SDE representing an Ornstein-Uhlenbeck process[6]

$$d\mathbf{u}(t) = -\Lambda\mathbf{u}(t)dt + \mathbf{B}d\mathbf{W}(t), \tag{57}$$

where $\mathbf{u}(t) = [x(t), p(t)]^\mathsf{T}$, and

$$\Lambda = \begin{bmatrix} 2\Lambda_\gamma & -1/m \\ m\omega^2 & 2\Lambda_\varepsilon \end{bmatrix}, \quad \mathbf{B} = \begin{bmatrix} \sqrt{\frac{\mathcal{D}_\gamma}{m^2\omega^4}} & 0 \\ 0 & \sqrt{m^2\mathcal{D}_\varepsilon} \end{bmatrix}, \tag{58}$$

are the drift and diffusion matrices respectively, and $d\mathbf{W}(t) = [dW_1(t), dW_2(t)]^\mathsf{T}$ is a vector of independent real Wiener processes with correlations $\langle dW_n(t)dW_m(t)\rangle = \delta_{mn}dt$. The SDE has the formal solution

$$\mathbf{u}(t) = \exp[-\Lambda t]\mathbf{u}(0) + \int_0^t \exp[-\Lambda(t - t')]\mathbf{B}d\mathbf{W}(t'), \tag{59}$$

where have assumed the initial state $\mathbf{u}(0)$ to be deterministic. The mean undergoes exponential decay $\langle \mathbf{u}(t)\rangle = \exp[-\Lambda t]\mathbf{u}(0)$.

---

[6]The defining property of an Ornstein-Uhlenbeck process is that the drift and diffusion matrices are independent of the stochastic field. Since the noise is additive, the distinction between Ito and Stratonovich is irrelevant for Ornstein-Uhlenbeck processes.

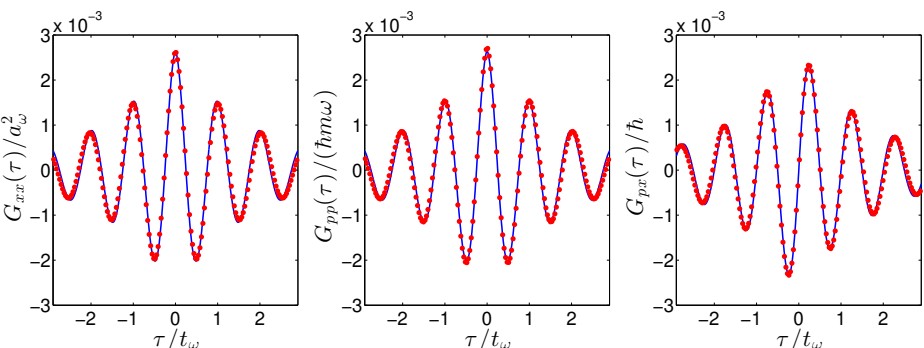

Figure 2: Steady-state time correlations for position-position (left), momentum-momentum (middle) and momentum-position (right), as determined by numerical solutions of the SPGPE (red dots) and the analytic solutions (solid blue) Eq. (63a)-Eq. (63c).

For a system with $x(0) = x_0$ and $p(0) = 0$, the center of mass position over time is given by

$$\langle x(t) \rangle = x_0 e^{-(\Lambda_\gamma + \Lambda_\varepsilon)t} \left[ \cos\left(\omega_{\gamma\varepsilon} t\right) + \frac{|\Lambda_\gamma - \Lambda_\varepsilon|}{\omega_{\gamma\varepsilon}} \sin\left(\omega_{\gamma\varepsilon} t\right) \right], \tag{60}$$

where we have defined the frequency $\omega_{\gamma\varepsilon} = \sqrt{\omega^2 - (\Lambda_\gamma - \Lambda_\varepsilon)^2}$. This simple result has a clear interpretation in terms of the decaying oscillatory motion of the center of mass with effective damping rates given by the SPGPE. Note that since we have arrived at a description with *additive* noise, the noise does not alter the average coordinate.

To study this regime of energy-damped dynamics numerically, we consider the steady-state correlations given by

$$\begin{aligned}
\mathbf{G}(\tau) &\equiv \lim_{t \to \infty} \langle [\mathbf{u}(t) - \langle \mathbf{u}(t) \rangle][\mathbf{u}(t+\tau) - \langle \mathbf{u}(t+\tau) \rangle]^{\mathsf{T}} \rangle \\
&= \int_{-\infty}^{\min(t, t+\tau)} \exp\left[-\mathbf{\Lambda}(t-t')\right] \mathbf{B}\mathbf{B}^{\mathsf{T}} \exp\left[-\mathbf{\Lambda}^{\mathsf{T}}(t+\tau-t')\right] dt'.
\end{aligned} \tag{61}$$

Fourier transforming with respect to $\tau$ in the steady-state, the fluctuation spectra are given by

$$\mathbf{S}(\Omega) = \frac{1}{2\pi}(\mathbf{\Lambda} + i\Omega)^{-1} \mathbf{B}\mathbf{B}^{\mathsf{T}}(\mathbf{\Lambda}^{\mathsf{T}} - i\Omega)^{-1}. \tag{62}$$

The steady-state correlations for position-position, momentum-momentum, and momentum-position are

$$G_{xx}(\tau) = -\frac{k_B T}{Nm} \frac{1}{\omega \omega_{\gamma\varepsilon}} e^{-(\Lambda_\gamma + \Lambda_\varepsilon)|\tau|} \sin\left(\omega_{\gamma\varepsilon}|\tau| - \sin^{-1}\left(\frac{\omega_{\gamma\varepsilon}}{\omega}\right)\right), \tag{63a}$$

$$G_{pp}(\tau) = \frac{mk_B T}{N} \frac{\omega}{\omega'_{\gamma\varepsilon}} e^{-(\Lambda_\gamma + \Lambda_\varepsilon)|\tau|} \sin\left(\omega_{\gamma\varepsilon}|\tau| + \sin^{-1}\left(\frac{\omega_{\gamma\varepsilon}}{\omega}\right)\right), \tag{63b}$$

$$G_{px}(\tau) = \frac{k_B T}{N} \frac{1}{\omega'_{\gamma\varepsilon}} e^{-(\Lambda_\gamma + \Lambda_\varepsilon)|\tau|} \sin\left(\omega'_{\gamma\varepsilon}\tau\right), \tag{63c}$$

respectively. The steady-state spectra for position-position, momentum-momentum, and momentum-

position are

$$S_{xx}(\Omega) = \frac{2k_{\mathrm{B}}T}{N\pi m\omega^2} \frac{\Lambda_{\varepsilon}(4\Lambda_{\gamma}\Lambda_{\varepsilon} + \omega^2) + \Lambda_{\gamma}\Omega^2}{(4\Lambda_{\gamma}\Lambda_{\varepsilon} + \omega^2)^2 - 2(\omega^2 - 2(\Lambda_{\varepsilon}^2 + \Lambda_{\gamma}^2))\Omega^2 + \Omega^4}, \tag{64a}$$

$$S_{pp}(\Omega) = \frac{2k_{\mathrm{B}}Tm}{N\pi} \frac{\Lambda_{\gamma}(4\Lambda_{\gamma}\Lambda_{\varepsilon} + \omega^2) + \Lambda_{\varepsilon}\Omega^2}{(4\Lambda_{\gamma}\Lambda_{\varepsilon} + \omega^2)^2 - 2(\omega^2 - 2(\Lambda_{\varepsilon}^2 + \Lambda_{\gamma}^2))\Omega^2 + \Omega^4}, \tag{64b}$$

$$S_{px}(\Omega) = \frac{2ik_{\mathrm{B}}T}{N\pi} \frac{(\Lambda_{\gamma} + \Lambda_{\varepsilon})\Omega}{(4\Lambda_{\gamma}\Lambda_{\varepsilon} + \omega^2)^2 - 2(\omega^2 - 2(\Lambda_{\varepsilon}^2 + \Lambda_{\gamma}^2))\Omega^2 + \Omega^4}, \tag{64c}$$

respectively.

We now test our analytical description against SPGPE simulations. While the SPGPE can be used for quantitative modelling [17, 29, 30] here our aim is simply to test our analytic solutions of the SPGPE using the SERs. We thus choose parameters that are representative of BEC experiments, and leave a first principles treatment of reservoir interactions [18] for future work.

We perform simulations of the 1D SPGPE [18] with the initial condition given by Eq. (53) with $x(0) = p(0) = 0$. We use a chemical potential of $\mu = 100\hbar\omega$, an energy cutoff of $\epsilon_{\mathrm{c}} = 250\hbar\omega$, a temperature of $T = 500\hbar\omega/k_{\mathrm{B}}$, an interaction strength of $g_{1\mathrm{D}} = 0.01\hbar\omega a_{\omega}$, an energy-damping rate of $\mathcal{M} = 0.0005a_{\omega}^2$, and a number-damping rate of $\gamma = 0.001$. We chose values for the damping rates such that $\Lambda_{\varepsilon} \approx \Lambda_{\gamma}$ and thus neither damping process is dominant over the other. Timescales are considered in units of the harmonic oscillator time period $t_{\omega} \equiv 2\pi/\omega$.

We compare our analytic solutions with the numerical data from equilibrated SPGPE simulations. In Fig. 2 we show the steady-state correlation functions for an ensemble of 5000 trajectories[7]. We see that the analytic and numeric results show excellent agreement for short times with differences becoming more pronounced for larger $\tau$. Similarly, the steady-state spectra for position-position, momentum-momentum, and momentum-position are shown in Fig. 3, where the numeric spectra are obtained using the Wiener-Khinchin theorem applied to the numeric steady-state correlations. The bulk oscillation seen in Fig. 2 is the Kohn mode, as seen from the peak at $\Omega \equiv \omega$ in Fig. 3. The spectral linewidth represents the decay time of the two-time correlation function, determined by $\Lambda_{\varepsilon}$ and $\Lambda_{\gamma}$, the energy and number damping rates. Again we see that the analytic and numeric results show good agreement. To assess the validity of neglecting projector terms in our analytical treatment, we evaluate their contribution numerically in Appendix B.3, finding that indeed the projector correction is negligible.

# 5 Discussion and Conclusions

## 5.1 Discussion

The SERs derived here using projected functional calculus contain many projector terms, posing a technical challenge. As an application of projected functional calculus, our approach bears comparison with that of Opanchuk *et al* [31], where a number of useful functional relations were derived for Wigner phase-space methods [6] involving projected fields. The action of the projector was reduced to the action of matrix operations, as is always possible in a finite basis. While not without its own technical challenges, our approach has the advantage that

---

[7]When calculating the correlations from the numeric data, we have assumed that the system has reached equilibrium after five trap cycles $t = 5t_{\omega}$, and used ergodic averaging over the remaining time interval $t = 5t_{\omega}$. With respect to the dissipation timescale $(\Lambda_{\gamma} + \Lambda_{\varepsilon})^{-1}$, the timescale of averaging is equivalent to $t = 2.78(\Lambda_{\gamma} + \Lambda_{\varepsilon})$.

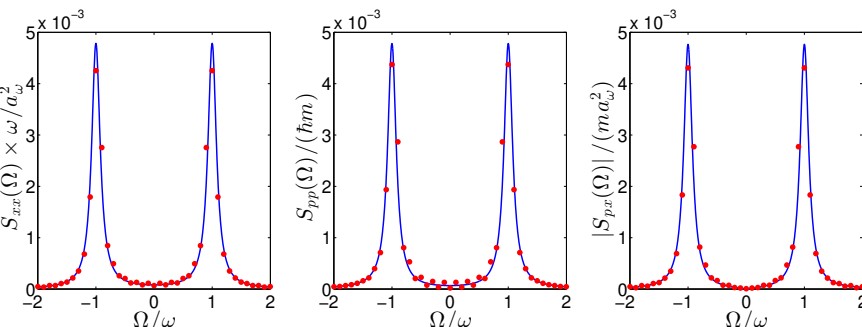

Figure 3: Steady-state spectra for position-position (left), momentum-momentum (middle) and momentum-position (right), as determined by numerical solutions of the SPGPE (red dots) and the analytic solutions (solid blue) Eq. (64a)-Eq. (64c).

the equations of motion have an obvious link with the continuum limit recovered for a high cutoff.

An alternate stochastic reservoir theory, namely the Stoof SGPE [32], has been used in several studies of finite-temperature BEC dynamics [33–37]. While the SGPE lacks a projection operator of the form used in the present work, our aims are similar in spirit to the work of Duine *et al.* [38] applying path-integral techniques to the Stoof SGPE to find effective stochastic equations for a reduced set of variables. However, the explicit high-energy cutoff in the SPGPE necessitates a different technical approach, with significant differences appearing in the resulting stochastic equations. As the energy-damping mechanism is absent from the Stoof SGPE, an interesting future direction is to identify further experimentally accessible regimes capable of distinguishing between the two approaches [30]. A related question receiving recent attention is that of ensemble equivalence [27]; in general the equilibrium ensemble generated by the SPGPE will be affected by the relative strengths of number and energy damping processes.

In applications of the SPGPE, the energy cutoff must be chosen such that the bulk of the coherent region modes are significantly occupied compared to the modes in the incoherent region. The relative occupation at the cutoff thus determines the presence of spurious dynamics due to the projector, and this population can typically be chosen to be of order unity. Ultimately, large projector corrections generate spurious dynamics and for a consistent treatment the cutoff should be chosen such that the projector corrections are small.

## 5.2 Conclusions

We have developed a set of exact stochastic Ehrenfest relations (SERs) for the complete stochastic projected Gross-Pitaevskii equation [10], an equation of motion which has significant application in the study of finite-temperature Bose-Einstein condensates [3, 9, 11]. In addition to the number-damping process, the SPGPE contains a number-conserving dissipative mechanism that transfers energy between the system and reservoir. Our main result, the SERs, retain the stochastic nature of the SPGPE and explicitly contain terms that result from both number- and energy-damping processes.

We tested our stochastic Ehrenfest equations in two ways. Considering the center of mass motion of a finite-temperature quasi-1D condensate near equilibrium, we tracked the size of the largest projector corrections and saw they are indeed small. We also compared the steady-state correlations of position and momentum to analytic solutions derived by neglecting the projector corrections, finding excellent agreement. Our chosen test system has the weakness that the center of mass motion in a purely harmonic trap is not strictly amenable to the reservoir theory due to a violation of Kohn's theorem, However, our treatment is physically relevant for

non-harmonic trapping, multicomponent systems, and other systems of interest that physically violate Kohn's theorem, provided a low-energy fraction is harmonically trapped. Indeed, since the thermal equilibrium properties involve small excursions from equilibrium, the confining potential is only required to be *locally* harmonic near the trap minimum and any number of non-harmonic effects may intrude at larger distances. The center of mass motion thus provides an excellent formal and numerical test of the SERs, being one of the simplest states of motion to handle analytically.

We have shown that SERs can be used to obtain analytic equations that agree with numerical solutions of the full SPGPE and offer some physical insight into the open system dynamics. Future work will explore systems involving analytically tractable excitations such as vortex decay in hard-wall confinement [46], soliton [39] and phase-slip dynamics [40] in toroidal confinement, sympathetic cooling [41,42], spinor BECs [43], and quantum turbulence in non-harmonic traps [44–47].

# Acknowledgements

We thank Matthew Reeves and Sam Rooney for valuable discussions.

**Funding information**   ASB was supported by the Marsden Fund (Contract UOO1726), and the Dodd-Walls Centre for Photonic and Quantum Technologies.

# A   Projector terms

## A.1   General operators

For general operator $\hat{A}$, the drift projector terms are

$$Q_A^H = -\frac{2}{\hbar}\text{Im}\int d^3\mathbf{r}\,(gn(\mathbf{r}) + \delta V(\mathbf{r},t))\psi(\mathbf{r})\mathcal{Q}^*\!\left(\frac{\delta A[\psi,\psi^*,t]}{\delta\psi(\mathbf{r})}\right)dt, \tag{65a}$$

$$Q_A^\gamma = \frac{2\gamma}{\hbar}\text{Re}\int d^3\mathbf{r}\,(gn(\mathbf{r}) + \delta V(\mathbf{r},t))\psi(\mathbf{r})\mathcal{Q}^*\!\left(\frac{\delta A[\psi,\psi^*,t]}{\delta\psi(\mathbf{r})}\right)dt, \tag{65b}$$

$$Q_A^\varepsilon = 2\mathcal{M}\,\text{Im}\int d^3\mathbf{k}\,S(k)\mathcal{F}\!\left[\psi^*(\mathbf{r})\mathcal{Q}\!\left(\frac{\delta A[\psi,\psi^*,t]}{\delta\psi^*(\mathbf{r})}\right)\right]^*\mathcal{F}[\nabla\cdot\mathbf{j}(\mathbf{r})]\,dt, \tag{65c}$$

the number-damping, and energy-damping diffusion projector terms are

$$D_A^\gamma = -\frac{4\gamma k_\text{B}T}{\hbar}\text{Re}\int d^3\mathbf{r}\,\frac{\delta A[\psi,\psi^*,t]}{\delta\psi^*(\mathbf{r})}\mathcal{Q}^*\!\left(\frac{\delta A[\psi,\psi^*,t]}{\delta\psi(\mathbf{r})}\right), \tag{66a}$$

$$\begin{aligned}
D_A^\varepsilon = {}&\frac{8\mathcal{M}k_\text{B}T}{\hbar}dt\int d^3\mathbf{k}\,S(k)\left|\mathcal{F}\!\left[\text{Im}\,\psi(\mathbf{r})\mathcal{Q}^*\!\left(\frac{\delta A[\psi,\psi^*,t]}{\delta\psi(\mathbf{r})}\right)\right]\right|^2 \\
&- \frac{16\mathcal{M}k_\text{B}T}{\hbar}dt\int d^3\mathbf{k}\,S(k)\mathcal{F}\!\left[\text{Im}\,\frac{\delta A[\psi,\psi^*,t]}{\delta\psi(\mathbf{r})}\psi(\mathbf{r})\right]^*\mathcal{F}\!\left[\text{Im}\,\psi(\mathbf{r})\mathcal{Q}^*\!\left(\frac{\delta A[\psi,\psi^*,t]}{\delta\psi(\mathbf{r})}\right)\right],
\end{aligned} \tag{66b}$$

and the epsilon term is

$$dA^\varepsilon = -\frac{k_\mathrm{B}T}{\hbar}\int d^3\mathbf{r}\int d^3\mathbf{r}'\left[\frac{\bar{\delta}A[\psi,\psi^*,t]}{\bar{\delta}\psi(\mathbf{r})}\delta(\mathbf{r},\mathbf{r}')\psi(\mathbf{r}')+\mathrm{h.c.}\right]\varepsilon(\mathbf{r}-\mathbf{r}')dt$$
$$+\frac{k_\mathrm{B}T}{\hbar}\int d^3\mathbf{r}\int d^3\mathbf{r}'\left[\frac{\bar{\delta}^{(2)}A[\psi,\psi^*,t]}{\bar{\delta}\psi^*(\mathbf{r}')\bar{\delta}\psi(\mathbf{r})}\psi^*(\mathbf{r}')\psi(\mathbf{r})+\mathrm{h.c.}\right]\varepsilon(\mathbf{r}-\mathbf{r}')dt$$
$$-\frac{k_\mathrm{B}T}{\hbar}\int d^3\mathbf{r}\int d^3\mathbf{r}'\left[\frac{\bar{\delta}^{(2)}A[\psi,\psi^*,t]}{\bar{\delta}\psi(\mathbf{r}')\bar{\delta}\psi(\mathbf{r})}\psi(\mathbf{r}')\psi(\mathbf{r})+\mathrm{h.c.}\right]\varepsilon(\mathbf{r}-\mathbf{r}')dt. \tag{67}$$

## A.2 One-body operators

For the special case where $\hat{A}$ is a one-body operator, the projector corrections take a simpler form. In this case we find the drift projector terms

$$Q_A^H = -\frac{2}{\hbar}\mathrm{Im}\int d^3\mathbf{r}\langle\psi|\hat{A}\hat{\mathcal{Q}}|\mathbf{r}\rangle(gn(\mathbf{r})+\delta V(\mathbf{r},t))\psi(\mathbf{r})dt \tag{68a}$$

$$Q_A^\gamma = \frac{2\gamma}{\hbar}\mathrm{Re}\int d^3\mathbf{r}\langle\psi|\hat{A}\hat{\mathcal{Q}}|\mathbf{r}\rangle(gn(\mathbf{r})+\delta V(\mathbf{r},t))\psi(\mathbf{r})dt \tag{68b}$$

$$Q_A^\varepsilon = 2\mathcal{M}\,\mathrm{Im}\int d^3\mathbf{k}\,S(k)\mathcal{F}\left[\langle\mathbf{r}|\hat{\mathcal{Q}}\hat{A}|\psi\rangle\psi^*(\mathbf{r})\right]^*\mathcal{F}[\nabla\cdot\mathbf{j}(\mathbf{r})]\,dt, \tag{68c}$$

diffusion projector terms

$$D_A^\gamma = -\frac{4\gamma k_\mathrm{B}T}{\hbar}\langle\hat{A}\hat{\mathcal{Q}}\hat{A}\rangle \tag{69}$$

$$D_A^\varepsilon = \frac{8\mathcal{M}k_\mathrm{B}T}{\hbar}dt\int d^3\mathbf{k}\,S(k)\left|\mathcal{F}\left[\mathrm{Im}\langle\psi|\hat{A}\hat{\mathcal{Q}}|\mathbf{r}\rangle\psi(\mathbf{r})\right]\right|^2$$
$$-\frac{16\mathcal{M}k_\mathrm{B}T}{\hbar}dt\int d^3\mathbf{k}\,S(k)\mathcal{F}\left[\mathrm{Im}\langle\psi|\hat{A}|\mathbf{r}\rangle\psi(\mathbf{r})\right]^*\mathcal{F}\left[\mathrm{Im}\langle\psi|\hat{A}\hat{\mathcal{Q}}|\mathbf{r}\rangle\psi(\mathbf{r})\right], \tag{70}$$

and the epsilon term

$$dA^\varepsilon = -\frac{k_\mathrm{B}T}{\hbar}dt\int d^3\mathbf{r}\int d^3\mathbf{r}'\langle\psi|\hat{A}\hat{\mathcal{P}}|\mathbf{r}\rangle\delta(\mathbf{r},\mathbf{r}')\psi(\mathbf{r}')\varepsilon(\mathbf{r}-\mathbf{r}')+\mathrm{h.c.}$$
$$+\frac{k_\mathrm{B}T}{\hbar}dt\int d^3\mathbf{r}\int d^3\mathbf{r}'\langle\mathbf{r}|\hat{\mathcal{P}}\hat{A}\hat{\mathcal{P}}|\mathbf{r}'\rangle\psi^*(\mathbf{r})\psi(\mathbf{r}')\varepsilon(\mathbf{r}-\mathbf{r}')+\mathrm{h.c.}. \tag{71}$$

# B  Centre of mass equations

Integrating over the Thomas-Fermi ansatz, the SERs for position and momentum take the form of a pair of coupled stochastic differential equations for the center of mass position and momentum

$$(\mathrm{I})dx(t) = \frac{1}{m}p(t)dt - \frac{\gamma m\omega^2}{\hbar}x(t)^3 dt - 2\Lambda_\gamma x(t)dt + dW_\gamma^x(t)$$
$$+ dW_\varepsilon^x(t) + q_x^H + q_x^\gamma + q_x^\varepsilon + dx^\varepsilon dt, \tag{72}$$

$$(\mathrm{I})dp(t) = -m\omega^2 x(t)dt - \frac{\gamma m\omega^2}{\hbar}x(t)^2 p(t)dt - 2\Lambda_\varepsilon p(t)dt + dW_\varepsilon^p(t)$$
$$+ dW_\gamma^p(t) + q_p^H + q_p^\gamma + q_p^\varepsilon + dp^\varepsilon dt. \tag{73}$$

Here the noise correlations are

$$\langle dW_\gamma^x(t)dW_\gamma^x(t)\rangle = \left(\frac{\mathcal{D}_\gamma}{m^2\omega^4} + \frac{4\gamma k_B T x(t)^2}{N\hbar} + d_x^\gamma\right)dt, \tag{74a}$$

$$\langle dW_\gamma^p(t)dW_\gamma^p(t)\rangle = d_p^\gamma dt, \tag{74b}$$

$$\langle dW_\gamma^x(t)dW_\gamma^p(t)\rangle = -\frac{4\mu_1\gamma k_B T}{g_1\hbar}p(t)x(t)dt, \tag{74c}$$

$$\langle dW_\varepsilon^x(t)dW_\varepsilon^x(t)\rangle = d_x^\varepsilon dt, \tag{74d}$$

$$\langle dW_\varepsilon^p(t)dW_\varepsilon^p(t)\rangle = \left(m^2\mathcal{D}_\varepsilon + d_p^\varepsilon\right)dt, \tag{74e}$$

$$\langle dW_\varepsilon^x(t)dW_\varepsilon^p(t)\rangle = d_{x,p}^\varepsilon dt, \tag{74f}$$

and the drift and diffusion rates are

$$\Lambda_\gamma = \frac{2\gamma\mu}{5\hbar}, \quad \mathcal{D}_\gamma = \frac{4m\omega^2 k_B T}{N}\Lambda_\gamma, \quad \mathcal{D}_\varepsilon = \frac{4k_B T}{mN_{\text{TF}}}\Lambda_\varepsilon,$$

$$\Lambda_\varepsilon = \frac{3\omega\mathcal{M}\hbar}{2gRa_\perp}\sqrt{\frac{\mu}{m\pi^3}}\int_0^\infty dq\, \text{erfcx}\left(\frac{|q|a_\perp}{R\sqrt{2}}\right)\frac{(\sin(q)-q\cos(q))^2}{q^4}, \tag{75}$$

derived from the effective one dimensional SPGPE [18] by integrating over the Thomas-Fermi ansatz. In terms of the dimensionless phase-space amplitude Eq. (54) the equation of motion becomes

$$(\mathbf{I})dz(t) = -i\omega z(t)dt - \left(\Lambda_\gamma + \Lambda_\varepsilon\right)z(t)dt - \left(\Lambda_\gamma - \Lambda_\varepsilon\right)z^*(t)dt$$
$$+ dW_{\gamma,1}^z(t) + dW_{\varepsilon,1}^z(t) + dW_{\gamma,2}^z(t) + dW_{\varepsilon,2}^z(t)$$
$$+ q_z^H dt + q_z^\gamma dt + q_z^\varepsilon dt + dz^\varepsilon dt, \tag{76}$$

where the noises have non-zero correlations

$$\langle dW_{\gamma,1}^z(t)dW_{\gamma,1}^z(t)\rangle = \frac{m\omega}{2\hbar}\left(\frac{\mathcal{D}_\gamma}{m^2\omega^4} + d_x^\gamma\right)dt, \tag{77a}$$

$$\langle dW_{\gamma,2}^z(t)dW_{\gamma,2}^z(t)\rangle = -\frac{1}{2\hbar m\omega}d_p^\gamma dt, \tag{77b}$$

$$\langle dW_{\varepsilon,1}^z(t)dW_{\varepsilon,1}^z(t)\rangle = \frac{m\omega}{2\hbar}d_x^\varepsilon dt, \tag{77c}$$

$$\langle dW_{\varepsilon,2}^z(t)dW_{\varepsilon 2}^z(t)\rangle = -\frac{1}{2\hbar m\omega}\left(m^2\mathcal{D}_\varepsilon + d_p^\varepsilon\right)dt, \tag{77d}$$

$$\langle dW_{\varepsilon,1}^z(t)dW_{\varepsilon 2}^z(t)\rangle = \frac{i}{2\hbar}d_{x,p}^\varepsilon dt, \tag{77e}$$

and the projector terms are given below. The projector terms are relatively simple for this SPGPE describing a BEC in an oblate parabolic trap with only one effective $C$-region dimension [18]. We first consider the position and momentum terms separately, and then give the terms for the dimensionless complex variable Eq. (54).

## B.1 Position and momentum

Noting that

$$\langle\psi|\hat{x}\hat{\mathcal{Q}}|x\rangle = \sqrt{\frac{\hbar(n_c+1)}{2m\omega}}\alpha_{n_c}^*(t)\phi_{n_c+1}^*(x), \tag{78}$$

$$\langle\psi|\hat{p}\hat{\mathcal{Q}}|x\rangle = -i\sqrt{\frac{\hbar m\omega(n_c+1)}{2}}\alpha_{n_c}^*(t)\phi_{n_c+1}^*(x), \tag{79}$$

the projector and epsilon terms for $x$ and $p$ are

$$q_x^H = -\frac{g_1}{N}\sqrt{\frac{2(n_c+1)}{\hbar m\omega}}\text{Im}\left\{\alpha_{n_c}^*(t)\int dx\,\phi_{n_c+1}^*(x)\psi(x)n(x)\right\}, \tag{80a}$$

$$q_x^\gamma = \frac{g_1\gamma}{N}\sqrt{\frac{2(n_c+1)}{\hbar m\omega}}\text{Re}\left\{\alpha_{n_c}^*(t)\int dx\,\phi_{n_c+1}^*(x)\psi(x)n(x)\right\}, \tag{80b}$$

$$q_x^\varepsilon = \frac{\mathcal{M}}{N}\sqrt{\frac{2\hbar(n_c+1)}{m\omega}}\int dk\,S_1(k)\mathcal{F}\left[\text{Im}\left\{\alpha_{n_c}^*(t)\phi_{n_c+1}^*(x)\psi(x)\right\}\right]^*\mathcal{F}[\partial_x j(x)], \tag{80c}$$

$$d_x^\gamma = -\frac{2\gamma k_B T(n_c+1)}{N^2 m\omega}\left|\alpha_{n_c}(t)\right|^2, \tag{80d}$$

$$d_x^\varepsilon = \frac{4\mathcal{M}k_B T(n_c+1)}{N^2 m\omega}\int dk\,S_1(k)\left|\mathcal{F}\left[\text{Im}\left\{\alpha_{n_c}^*(t)\phi_{n_c+1}^*(x)\psi(x)\right\}\right]\right|^2, \tag{80e}$$

$$\begin{aligned}
dx^\varepsilon = &-\frac{k_B T}{N\hbar}\int dx\int dx'\langle\psi|\hat{x}\hat{\mathcal{P}}|x\rangle\delta(x,x')\psi(x')\varepsilon(x-x')+\text{h.c.}\\
&+\frac{k_B T}{N\hbar}\int dx\int dx'\langle x|\hat{\mathcal{P}}\hat{x}\hat{\mathcal{P}}|x'\rangle\psi^*(x)\psi(x')\varepsilon(x-x')+\text{h.c.},
\end{aligned} \tag{80f}$$

and

$$q_p^H = \frac{g_1}{N}\sqrt{\frac{2m\omega(n_c+1)}{\hbar}}\text{Re}\left\{\alpha_{n_c}^*(t)\int dx\,\phi_{n_c+1}^*(x)\psi(x)n(x)\right\}, \tag{81a}$$

$$q_p^\gamma = \frac{g_1\gamma}{N}\sqrt{\frac{2m\omega(n_c+1)}{\hbar}}\text{Im}\left\{\alpha_{n_c}^*(t)\int dx\,\phi_{n_c+1}^*(x)\psi(x)n(x)\right\}, \tag{81b}$$

$$q_p^\varepsilon = \frac{\mathcal{M}}{N}\sqrt{2\hbar m\omega(n_c+1)}\int dk\,S_1(k)\mathcal{F}\left[\text{Re}\left\{\alpha_{n_c}^*(t)\phi_{n_c+1}^*(x)\psi(x)\right\}\right]^*\mathcal{F}[\partial_x j(x)], \tag{81c}$$

$$d_p^\gamma = -\frac{2\gamma m\omega k_B T(n_c+1)}{N^2}\left|\alpha_{n_c}(t)\right|^2, \tag{81d}$$

$$\begin{aligned}
d_p^\varepsilon = \frac{4\mathcal{M}k_B T}{N^2}\int dk\,S_1(k)\Bigg(&m\omega(n_c+1)\left|\mathcal{F}\left[\text{Re}\left\{\alpha_{n_c}^*(t)\phi_{n_c+1}^*(x)\psi(x)\right\}\right]\right|^2\\
&-\sqrt{8\hbar m\omega(n_c+1)}\text{Re}\left\{\mathcal{F}\left[\text{Re}\left\{\alpha_{n_c}^*(t)\phi_{n_c+1}^*(x)\psi(x)\right\}\right]\mathcal{F}[\partial_x n(x)]^*\right\}\Bigg), \tag{81e}
\end{aligned}$$

$$\begin{aligned}
dp^\varepsilon = &-\frac{k_B T}{N\hbar}\int dx\int dx'\langle\psi|\hat{p}\hat{\mathcal{P}}|x\rangle\delta(x,x')\psi(x')\varepsilon(x-x')+\text{h.c.}\\
&+\frac{k_B T}{N\hbar}\int dx\int dx'\langle x|\hat{\mathcal{P}}\hat{p}\hat{\mathcal{P}}|x'\rangle\psi^*(x)\psi(x')\varepsilon(x-x')+\text{h.c..}
\end{aligned} \tag{81f}$$

## B.2 Dimensionless variable $z(t)$

The projector corrections for the dimensionless variable phase-space amplitude $z(t)$ are

$$q_z^H = \frac{ig_1}{\hbar N} \sqrt{n_c + 1}\, \alpha_{n_c}^*(t) \int dx\, \phi_{n_c+1}^*(x) \psi(x) n(x), \tag{82a}$$

$$q_z^\gamma = \frac{\gamma g_1}{\hbar N} \sqrt{n_c + 1}\, \alpha_{n_c}^*(t) \int dx\, \phi_{n_c+1}^*(x) \psi(x) n(x), \tag{82b}$$

$$q_z^\varepsilon = \frac{i\mathcal{M}}{N} \sqrt{n_c + 1}\, \alpha_{n_c}^*(t) \int dk\, S_1(k) \mathcal{F}\left[\phi_{n_c+1}^*(x)\psi(x)\right]^* \mathcal{F}[\partial_x j(x)], \tag{82c}$$

$$d_z^\gamma = -\frac{\gamma k_B T (n_c + 1)}{\hbar N^2} \left|\alpha_{n_c}(t)\right|^2, \tag{82d}$$

$$d_z^{\varepsilon,a} = \frac{2\mathcal{M}k_B T (n_c + 1)}{\hbar N^2} \int dk\, S_1(k) \left|\mathcal{F}\left[\mathrm{Im}\left\{\alpha_{n_c}^*(t)\phi_{n_c+1}^*(x)\psi(x)\right\}\right]\right|^2, \tag{82e}$$

$$d_z^{\varepsilon,b} = \frac{2\mathcal{M}k_B T}{\hbar N^2} \int dk\, S_1(k) \Bigg( (n_c + 1) \left|\mathcal{F}\left[\mathrm{Re}\left\{\alpha_{n_c}^*(t)\phi_{n_c+1}^*(x)\psi(x)\right\}\right]\right|^2$$

$$+ \sqrt{\frac{8\hbar(n_c + 1)}{m\omega}} \mathrm{Re}\left\{\mathcal{F}\left[\mathrm{Re}\left\{\alpha_{n_c}^*(t)\phi_{n_c+1}^*(x)\psi(x)\right\}\right]^* \mathcal{F}[\partial_x n(x)]\right\} \Bigg), \tag{82f}$$

$$dz^\varepsilon = -\frac{k_B T}{N\hbar} \int dx \int dx' \langle \psi|\hat{z}\hat{\mathcal{P}}|x\rangle \delta(x,x')\psi(x')\varepsilon(x-x') + \mathrm{h.c.}$$

$$+ \frac{k_B T}{N\hbar} \int dx \int dx' \langle x|\hat{\mathcal{P}}\hat{z}\hat{\mathcal{P}}|x'\rangle \psi^*(x)\psi(x')\varepsilon(x-x') + \mathrm{h.c.} \tag{82g}$$

## B.3 Numerical evaluation of projector terms

With the exception of the epsilon correction Eq. (82g), all the cutoff terms are a result of mode mixing between the highest energy coherent mode and the lowest energy incoherent mode. In the limit of imposing a very high energy cutoff, the populations of the modes above the cutoff approach zero and hence all the cutoff terms go to zero. For a well-chosen but finite cutoff the integrals involving the overlap of the lowest energy incoherent mode $\phi_{n_c+1}(x)$ and the coherent field wave function $\psi(x)$ should be small, as $\phi_{n_c+1}(x)$ is highly oscillatory and the mode population is also small by definition. We claim that for a well-chosen cutoff the cutoff terms are small enough such that they may be neglected, and we justify this in two ways. Firstly, we consider the magnitude of a selection of the cutoff terms by calculating them numerically. Second, we consider the analytic solutions that can be found by neglecting the cutoff terms and show that these agree well with simulations of the 1D SPGPE.

When considering the cutoff terms involving mode mixing at the cutoff (i.e. all except the epsilon correction), we note that only the Hamiltonian cutoff term $q_z^H$ Eq. (82a) does not have one of the damping rates as a multiplying factor. As the damping rates have a typical value several orders of magnitude less than unity, it is reasonable to expect that of all these terms, the Hamiltonian cutoff term will be the largest. If we show that $q_z^H$ is small enough to be neglected then we can reason that $q_z^\gamma$ Eq. (82b), $q_z^\varepsilon$ Eq. (82c), $d_z^\gamma$ Eq. (82d), $d_z^\varepsilon|_{(1)}$ Eq. (82e), and $d_z^\varepsilon|_{(2)}$ Eq. (82f) are smaller still and so can certainly be neglected also.

The epsilon term $dz^\varepsilon$ Eq. (82g) is distinct from the other terms, as it is not a result of mode mixing at the cutoff. We expect the two terms in the epsilon term to almost cancel, as it is clear this is the case in the limit that the projector becomes the identity. This is an important result of the earlier step where we found that writing the SPGPE in Ito form resulted in an extra term; without this extra term the epsilon term would in general be non-negligible, and

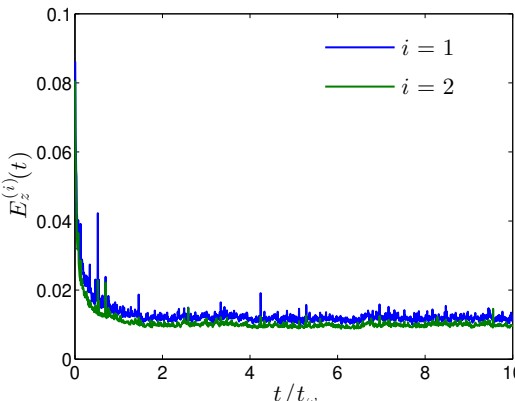

Figure 4: The mean relative cutoff term magnitudes, Eq. (83), for $q_z^H$ (blue) and $dz^\varepsilon$ (green) over time determined numerically for an ensemble of 1000 trajectories.

in fact diverges in the infinite cutoff limit. We thus monitor the magnitudes of $q_z^H$ and $dz^\varepsilon$ over the course of an ensemble of numerical trajectories.

We define the relative cutoff term magnitudes by

$$E_z^{(1)}(t) \;=\; \left| q_z^H(t)/\dot{z}(t) \right|, \quad E_z^{(2)}(t) = \left| dz^\varepsilon(t)/\dot{z}(t) \right|, \tag{83}$$

noting that $|\dot{z}(t)|$ is strictly non-zero for harmonic motion. If these values remain significantly less than unity, then we may conclude that the effects of the cutoff terms are negligible. For the simulations of section 4.2, example relative cutoff term magnitudes are shown in Fig. 4, where the initial state is Eq. (53) with $x(0) = p(0) = 0$ and we have taken an ensemble average over 1000 trajectories. While the relative magnitudes can reach as high as $\sim 0.1$ early in the dynamics, we see that once the system has equilibrated they approach a steady-state of order $\sim 0.01$.

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
