# Peer review of "Dynamics of Hot Bose-Einstein Condensates: stochastic Ehrenfest relations for number and energy damping"

_SciPost Physics, doi:SciPost Phys. 8, 029 (2020)_

## Round 1 · Referee Report · Anonymous (Referee 1) · 2019-9-25

Report

In the present paper the authors discuss Ehrenfest relations of the SPGPE. The latter is a well-established method to describe thermal effects in a BEC of ultracold atoms. In its present form I do not recommend publication of this paper for the following reasons:

1) The paper is way too long and should much more focus on the new results, i.e., the Ehrenfest relations, the authors have obtained. There existst already a large amount of literature on the SPGPE an his does not require to be reproduced here.

2) It is not fully clear to me to what equilibrium the noises that are used by the authors lead. I have the impression that the equilibrium is the classical equilibrium that is valid for the low-energy physics, but not the quantum equilibrium that is needed at high energies (the thermal cloud). I think that it is very important that the authors discuss this equilibrium of the SPGPE so that the readers can judge its validity. At present the authors do not discuss this point at all.

3) The use of a high-energy reservoir in the SPGPE, is reasonable if one is interested in condensate formation, for which this method was originally developed. However, the authors now apply this method for collective modes and in particular the Kohn mode. The results for the Kohn mode immediately show the draw-back of this high-energy reservoir, because the Kohn theorem is no longer satisfied and the Kohn modes becomes damped. This is a serious problem, because in typical ultracold atom experiments the Kohn theorem is actually satisfied and experimentalists actually use it to measure the trap frequencies. So this clearly seriously restricts the use of this method to situations where the Kohn theorem is violated, which is exceptional. Moreover, the Kohn theorem will then be violated for other reasons and not because of a high-energy reservoir. This therefore leads to the question what precisely the purpose of this paper is as the theory does not seem immediately applicable to real experiments.

---

## Round 1 · Referee Report · Anonymous (Referee 2) · 2019-10-22

Report

This is an interesting piece of work from one of the centres of finite-temperature BEC (and related many body system) theory. The work and conclusions are basically fully sound, and my comments are essentially to do with paying some more careful attention to how the problem is set up. On a more minor note there is the odd bit of careless spelling/grammar/formatting, and the huge chunk of white space at the bottom of page 11 is fairly unattractive! I will put my numbered comments int he next box below.

Requested changes

  1. Describing the ZNG approach as well-suited for the low-temperature regime; I think the authors need to be a bit more specific as to what they mean by this regime. I would say ZNG is well suited to regimes where there is both significant condensate and significant thermal cloud - I would not consider it to be, for example, especially well suited to describing the very low temperature regime where depletion from the condensate is due primarily to quantum fluctuations or dynamics, and my understanding is that it is quite well suited to describing e.g. excitation spectra at temperatures considered to be significant enough to have a noticeable effect - witness the early calculations of Jackson and Zaremba in describing the JILA experiment in the earlier days of BEC.

  2. "the expectation of momentum", e.g., to my mind makes little grammatical sense relative to "the expectation value of momentum"

  3. I don't think it is right to say "the Gross-Pitaevskii equation is obtained by taking the functional derivative of the Gross-Pitaevskii Hamiltonian". Something like "can be generated from" would be better.

  4. As a notational note, starting in equation 5, and very frequently thereafter, ket notation is used to describe what are effectively field modes. The authors may find this to be a convenience, however it should not be forgotten that these do not, in a direct sense, describe the state vector of the system. And to describe the state vector of the system in occupation number representation (as must necessarily be assumed when working in a many-body quantum-field-theory formalism), it is a ket notation that is customarily used. Although this is sort of obvious to someone relatively experienced, a certain lack of notational care of this kind can cause considerable confusion.

  5. I'm not sure that I'm happy with the definitions given of R, P and L. As described later they are effectively single-body quantities, i.e., R and P are the centre-of mass position and momentum of the many body system, and I'm tempted to suggest some defining equations to pin down exactly what these quantities are.

  6. A little bit more motivation as to why the basis (which is in principle arbitrary -- except for the fact that it is cut off) would be desirable (below Eq. (11))

  7. Analagous should be spelled analogous (below Eq. (17)).

  8. Below Eq. (23), I think the discussion of conservation laws could be clarified. The point seems to be that the way the projector is defined means that any conservation laws observed in the "total" state space/system must also be observed within the subspace described by C - therefore it is necessary, or at least highly desirable, that this should be respected in any numerical implementation

  9. In the same place: "quantities may exist" (not "exists").

  10. The subscript "dB" (for de Broglie), or anything else where the subscript is a word or an abbreviation of words) should really be formatted in roman (not maths italic) type. "1D" is another example)

  11. Eq. (31), what seem to be Fourier transforms are not defined as such.

  12. Re. Eq. (41): while I understand what the authors are trying to say, and think it basically sound, they are using a notation which looks in a sense self evidently like an expectation value for a single-body Schrodinger equation, when that's not really what is going on, due to the system under consideration being a many-body system. In first =-quantised form $\hat{A}=\sum{k=1}^{N}\hat{A}_{k}$ where the \hat{A}_{k} are operators corresponding to a particular observable for a particular particle. This is also a case where what are effectively field modes are used in a way which makes them look like basis states for a single-body system. The outcomes are not incorrect, but there is some potential for confusion.

---

## Round 2 · Referee Report · Anonymous · 2019-12-6

Report

I cannot recommend publication of this paper, because the authors have just not overcome my remarks in my previous report. 1) The paper is way too long, 2) the equilibrium is not properly discusses and if it is the classical (field) equilibrium that I think is the case here, then it is actually not appropriate for the normal state of the gas, which the authors are discussing, 3) the Kohn theorem is violated and this shows the theory is not appropriate for real experiments.

  • validity: poor
  • significance: good
  • originality: good
  • clarity: good
  • formatting: good
  • grammar: good

Author:  Ashton Bradley  on 2019-12-16  [id 686]

(in reply to Report 1 on 2019-12-06)
Category:
reply to objection

We first address objection

3) the Kohn theorem is violated and this shows the theory is not appropriate for real experiments

We have resubmitted v3 to the arXiv

https://arxiv.org/abs/1908.05809

with revisions to address the point raised, as detailed below.

%-------------------------------- Revisions:

  • Introduced acronym Stochastic Ehrenfest relation (SER)

  • Minor change of wording in second to last paragraph of introduction:

“As a test we apply the Ehrenfest relations to the centre of mass fluctuations of a harmonically trapped system tightly confined along two spatial dimensions. We find that the analtyic solution of the SER for the centre of mass is in close agreement with SPGPE simulations. ”

  • Replaced center —> centre throughout

  • Reordered the wording in the “v) Thermal equilibrium” paragraph at the end of section 3 on page 13, to improve clarity

  • Introduced acronym TIRA (Time Independent Reservoir Approximation)

  • Start of section 4: substantial revisions of the text to point out several systems of interest where Kohn’s theorem is not satisfied by the system, and hence the SPGPE reservoir theory provides a useful approximation.

We have also emphasized that the SPGPE has been used to give a first-principles treatment of high temperature non-equilibrium experiments. In particular, spontaneous vortex formation was described using one fitted parameter (ref [3] of revised manuscript), and the SPGPE gave a quantitative description of the experiment in ref [17] of the revised manuscript, with no fitted parameters.

"While the time-independent reservoir approximation (TIRA) is not strictly applicable for scalar BEC in the purely harmonic trap, there are a number of physical systems where it is applicable: Kohn’s theorem does not apply to a scalar Bose gas held in a harmonic trap if the trap becomes non harmonic at high energy. The theorem is also inapplicable for a harmonically trapped system if the reservoir consists of second atomic species confined by a different trapping potential, as may occur during sympathetic cooling. Furthermore, any system that is not harmonically trapped will not obey Kohn’s theorem, and is thus potentially amenable to the TIRA. Example systems in non-harmonic traps for which the theory is applicable include vortex decay in hard-wall confinement [46], soliton decay in a 1D toroidal trap (where the present approach was first used) [19], and persistent current formation in a 3D toroidal trap, where SPGPE simulations [17] compare well with experiment [44].

In this work our approach is simply to test the formalism on a simple model system within the TIRA by integrating out the spatial degrees of freedom to find effective stochastic equations of motion for the centre of mass. We stress that the TIRA approached used here is physically valid for (at least) two scenarios if immediate interest: non-harmonic trapping at high energies for a scalar BEC, and sympathetic cooling involving two BEC components in different harmonic traps [14]."

These points of context are also mentioned briefly in footnote 3 (page 14) of the revised manuscript.

  • Conclusions substantially revised to further clarify the physical applicability of the theory of the harmonically trapped system to real physical systems. In particular:

“We tested our stochastic Ehrenfest equations in two ways. Considering the centre of mass motion of a finite-temperature quasi-1D condensate near equilibrium, we tracked the size of the largest projector corrections and saw they are indeed small. We also compared the steady-state correlations of position and momentum to analytic solutions derived by neglecting the projector corrections, finding excellent agreement. Our chosen test system has the weakness that the centre of mass motion in a purely harmonic trap is not strictly amenable to the reservoir theory due to a violation of Kohn's theorem. However, our treatment is physically relevant for non-harmonic trapping, multicomponent systems, and other systems of interest that physically violate Kohn's theorem, provided a low-energy fraction is harmonically trapped. Indeed, since the thermal equilibrium properties involve small excursions from equilibrium, the confining potential is only required to be \emph{locally} harmonic near the trap minimum and any number of non-harmonic effects may intrude at larger distances. The centre of mass motion thus provides an excellent formal and numerical test of the SERs, being one of the simplest states of motion to handle analytically. \par We have shown that SERs can be used to obtain analytic equations that agree with numerical solutions of the full SPGPE and offer some physical insight into the open system dynamics. Future work will explore systems involving analytically tractable excitations such as vortex decay in hard-wall confinement [46], soliton [39] and phase-slip dynamics [40] in toroidal confinement, sympathetic cooling [41,42], spinor BECs [23], and quantum turbulence in non-harmonic confinement [44-47].” %--------------------------------------------------

In short, there are a number of systems under current and future study for which the theory is applicable. For our purpose it forms a very useful test for the formalism.

We hope that these changes are appropriate to address the remaining concerns regarding physical applicability.

Regarding Referee comment

2) the equilibrium is not properly discusses and if it is the classical (field) equilibrium that I think is the case here, then it is actually not appropriate for the normal state of the gas, which the authors are discussing

We find this comment unclear. We do not discuss the normal state of the gas directly, as our theory is a stochastic field theory of the low-energy partially degenerate fraction of the gas. There is a classical field equilibrium (within the truncated Wigner classical field approximation), that contains the condensate and a low energy normal fraction, but the latter must be extracted numerically (typically via Penrose-Onsager).

A high-energy normal fraction of the gas appears explicitly via the reservoir, due to the choice of the energy cutoff - it must be chosen to separate the coherent region from the incoherent region of phase space.

The normal fraction of the gas is thus included in the properties of the reservoir (gaussian statistics, chemical potential, temperature, and reservoir interaction rates), formulated using the single-particle Wigner function for the high energy part of the field.

In short, we are not sure we understand the question, but we would be happy to provide an answer to a clarified question.

With Best Regards,

Ashton Bradley (on behalf of the authors).

---

## Round 2 · Author Response

We thank the reviewers for their comments. While Referee A would like to see the article shortened and the application removed, Referee B did not raise an objection to the length, and is generally quite positive about the manuscript. In light of this, and because we believe that even a shortened application section makes the utility of the article that much clearer, we have instead tried to shorten the main text where possible by removing background, and moving technical calculations to appendices.

We thank both referees for their constructive feedback, and we feel that these and the other revisions have strengthened the paper. We believe we have addressed all of the comments, and hope that the manuscript is now suitable for publication.

Reviewer A

  1. "The paper is way too long and should much more focus on the new results, i.e., the Ehrenfest relations, the authors have obtained. There exists already a large amount of literature on the SPGPE and this does not require to be reproduced here. We have shortened the introduction by about 1 page by removing or shortening any features of the presentation that aren’t absolutely essential to the current manuscript."

We do not agree with the referee that we should only focus on the formalism here, as a primary aim is to demonstrate the utility of the formalism. However, we accept that the paper was long, and have rewritten the application section to focus more directly on the main results, moving some of the more technical developments into an appendix. The result is to give a more direct presentation of the main outcome of applying the formalism. We have also removed 2 figures that were not central to the paper, and shortened the background. We hope that these changes addresses the referee’s comment.

  1. "It is not fully clear to me to what equilibrium the noises that are used by the authors lead. I have the impression that the equilibrium is the classical equilibrium that is valid for the low-energy physics, but not the quantum equilibrium that is needed at high energies (the thermal cloud). I think that it is very important that the authors discuss this equilibrium of the SPGPE so that the readers can judge its validity. At present the authors do not discuss this point at all."

The equilibrium is a semi-classical (truncated Wigner, in which quantum noise is small compared to thermal noise) grand-canonical ensemble for the low-energy region (C-region). The C-region is described by a stochastic c-field, and this is the dynamical variable of the theory. The region of the gas that is above the energy cutoff, the I-region, is described semi-classically, but in a different approximate. The one-body Wigner function for the high-energy fraction of the gas is treated semiclassically, giving a quantum Boltzmann-like description for population dynamics (where phase coherences are absent). Typically the I-region is described as time independent, and thus is parameterised by temperature and chemical potential. We have expanded our discussion of equilibrium in the discussion of our stochastic Ehrenfest relations. We have also expanded upon our discussion of the role of the cutoff and choice of representation basis in determining equilibrium properties.

  1. "The use of a high-energy reservoir in the SPGPE, is reasonable if one is interested in condensate formation, for which this method was originally developed. However, the authors now apply this method for collective modes and in particular the Kohn mode. The results for the Kohn mode immediately show the draw-back of this high-energy reservoir, because the Kohn theorem is no longer satisfied and the Kohn modes becomes damped. This is a serious problem, because in typical ultracold atom experiments the Kohn theorem is actually satisfied and experimentalists actually use it to measure the trap frequencies. So this clearly seriously restricts the use of this method to situations where the Kohn theorem is violated, which is exceptional. Moreover, the Kohn theorem will then be violated for other reasons and not because of a high-energy reservoir. This therefore leads to the question what precisely the purpose of this paper is as the theory does not seem immediately applicable to real experiments."

We agree (and we said as much in the original manuscript) that this poses a problem for describing the harmonically trapped system for a single-component BEC. However, we do not agree that the violation of Kohn’s theorem poses a problem in general. There are many situations where Kohn’s theorem is not physically applicable. The approach we take in this work is relevant for such scenarios provided the high-energy reservoir is not significantly affected by the dynamics. The time-independent reservoir approximation is then appropriate. However, it is important to emphasize that in the present work our aim is to test the Ehrefest relations, rather than make concrete predictions for a specific system. We have expanded our discussion at the start of section 4 of the manuscript to better emphasise these points.

Reviewer B 1. "Describing the ZNG approach as well suited for the low-temperature regime; I think the authors need to be a bit more specific as to what they mean by this regime. I would say ZNG is well suited to regimes where there is both significant condensate and significant thermal cloud - I would not consider it to be, for example, especially well suited to describing the very low temperature regime where depletion from the condensate is due primarily to quantum fluctuations or dynamics, and my understanding is that it is quite well suited to describing e.g. excitation spectra at temperatures considered to be significant enough to have a noticeable effect - witness the early calculations of Jackson and Zaremba in describing the JILA experiment in the earlier days of BEC."

We have revised our discussion to refer to “significant non-condensate fraction”

We have also changed our description of the success of ZNG to include collective mode simulations (ref [5] of revised manuscript), which we agree are important to point out:

“However, it's strength for low temperatures presents a limitation at high temperatures: despite notable successes in, e.g. collective modes [5], the symmetry-breaking ansatz has limited scope for describing for strongly fluctuating systems containing large non-condensate fraction.”

  1. "the expectation of momentum", e.g., to my mind makes little grammatical sense relative to "the expectation value of momentum"

We have made the suggested change.

  1. "I don't think it is right to say "the Gross-Pitaevskii equation is obtained by taking the functional derivative of the Gross-Pitaevskii Hamiltonian". Something like "can be generated from" would be better."

Functional differentiation of the Hamiltonian is one way to obtain the GPE, using Hamilton’s equations of motion for the continuous classical field. We agree that this is not a “derivation”, but more a formal mathematical development. However, as we use this approach later for the projected classical field, we need to be clear that this is one way to develop it and link the two concepts. We have change the wording to:

“the Gross-Pitaevskii equation can be generated by taking the functional derivative of the Gross-Pitaevskii Hamiltonian”.

  1. "As a notational note, starting in equation 5, and very frequently thereafter, ket notation is used to describe what are effectively field modes. The authors may find this to be a convenience, however it should not be forgotten that these do not, in a direct sense, describe the state vector of the system. And to describe the state vector of the system in occupation number representation (as must necessarily be assumed when working in a many-body quantum-field-theory formalism), it is a ket notation that is customarily used. Although this is sort of obvious to someone relatively experienced, a certain lack of notational care of this kind can cause considerable confusion."

Whenever we use the state ket notation for $|\psi \rangle$ we are using this to describe the abstract state ket for a pure state associated with a particular realisation of the stochastic field. The state ket formalism is merely a convenient way to change bases for a particular wavefunction, and does not represent the many body quantum state which requires a density matrix. Thus the average $\langle \hat A \rangle$ only refers to the wavefunction average over a pure state, and not an expectation value for the many-body density matrix.

We are not working with an occupation number representation as we are using the Wigner phase space representation of the quantum field, which instead utilises the coherent state basis. Both bases are complete for the many body problem, but the latter is more suited for bosonic systems exhibiting matter-wave coherence. This comment is helpful in that it gives a chance to clarify the notation, which we agree can be confusing.

We have added a discussion in the text to clarify this point, at the end of section 2.1:

“As is often convenient when changing representations, here we are using $|\psi \rangle$ to represent the abstract state ket associated with the pure-state wavefunction $\psi(\mathbf{r})=\langle\mathbf{r}|\psi\rangle$. In the context of phase-space methods introduced below, it should be noted that we do not use this notation for the many-body quantum state, but just as a convenient shorthand for the wavefunction appearing in individual trajectories. Many-body expectation values are calculated as ensemble averages over trajectories of the stochastic field theory [9]. When generalizing the above definitions [(6)-(10)] to the many-body theory as described below, the average $\langle \cdot\rangle$ refers only to the wavefunction expectation over position coordinates. The combination of wavefunction and ensemble average will be denoted by $\langle\langle \cdot \rangle\rangle $.”

  1. "I'm not sure that I'm happy with the definitions given of R, P and L. As described later they are effectively single-body quantities, i.e., R and P are the centre-of mass position and momentum of the many body system, and I'm tempted to suggest some defining equations to pin down exactly what these quantities are."

We now give a concrete example for R, and the others follow analogously. This should also be clarified by our response to the previous point.

  1. "A little bit more motivation as to why the basis (which is in principle arbitrary -- except for the fact that it is cut off) would be desirable (below Eq. (11))"

We have added subsection 2.2.3 which discusses the role of the basis, and cutoff.

  1. "Analagous should be spelled analogous (below Eq. (17))."

Fixed.

  1. "Below Eq. (23), I think the discussion of conservation laws could be clarified. The point seems to be that the way the projector is defined means that any conservation laws observed in the "total" state space/system must also be observed within the subspace described by C - therefore it is necessary, or at least highly desirable, that this should be respected in any numerical implementation"

We have clarified this by slightly expanding the discussion below equation (22) of the revised manuscript.

  1. "In the same place: "quantities may exist" (not "exists")."

Fixed.

  1. "The subscript "dB" (for de Broglie), or anything else where the subscript is a word or an abbreviation of words) should really be formatted in roman (not maths italic) type. "1D" is another example)"

Fixed.

  1. "Eq. (31), what seem to be Fourier transforms are not defined as such."

We have now given a definition, Eq. (35) of the revised manuscript.

  1. "Re. Eq. (41): while I understand what the authors are trying to say, and think it basically sound, they are using a notation which looks in a sense self evidently like an expectation value for a single-body Schrodinger equation, when that's not really what is going on, due to the system under consideration being a many-body system. In first =-quantised form ^A=∑k=1N^Ak where the \hat{A}_{k} are operators corresponding to a particular observable for a particular particle. This is also a case where what are effectively field modes are used in a way which makes them look like basis states for a single-body system. The outcomes are not incorrect, but there is some potential for confusion."

We agree that this was potentially confusing and have changed the wording in section 3.3 to read:

“Consider a quantity that is the pure-state trace of a one-body operator \ali{A[\psi,\psi^]&=\bra{\psi} \hat{A} \ket{\psi}=\int d^3\rangle\rangle \bra{\psi} \hat{A} \ket{\rangle\rangle}\psi(\mathbf{r}\mathbf{r}),} where the operator $\hat A$ is independent of $\psi, \psi^$. We emphasize that this quantity is the spatial average for a pure state wavefunction associated with each trajectory, and does not represent the many-body expectation value found by tracing over the density matrix (the ensemble average).”

---

## Round 2 · List of Changes

1. Shortened background by removing redundant material.

2. Removed Figures 2 and 3 from the manuscript. We felt that these took up a lot of space, and did not contribute much to the main point of the article.

3. We now include a short discussion of equilibrium in section 3.5

4. Just before Eq (12), clarified that the basis is a single-particle basis.

5. Moved background material on the Ito form of the SPGPE into Section 3.

6. Moved technical details about finding the effective equation of motion from Section 4 into Appendix B.

7. Added reference [5] Jackson and Zaremba, Phys. Rev. Lett. 88(18), 180402 (2002)

8. Added reference [28] Pietraszewicz and Deuar, Phys. Rev. A 98(2), 023622 (2018)

9. Expanded discussion of ZNG in section 1:

“However, it's strength for low temperatures presents a limitation at high temperatures: despite notable successes in, e.g. collective modes [5], the symmetry-breaking ansatz has limited scope for describing for strongly fluctuating systems containing large non-condensate fraction.”

10. Added text to end of section 2.1:

“As is often convenient when changing representations, here we are using $|\psi\rangle$ to represent the abstract state ket associated with the pure-state wavefunction $\psi(\mathbf{r})=\langle \mathbf{r} |\psi\rangle$. In the context of phase-space methods introduced below, it should be noted that we do not use this notation for the many-body quantum state, but just as a convenient shorthand for the wavefunction appearing in individual trajectories. Many-body expectation values are calculated as ensemble averages over trajectories of the stochastic field theory [9]. When generalizing the above definitions [(\ref{rdef})---(\ref{ndef})] to the many-body theory as described below, the average $\langle \cdot\rangle$ refers only to the wavefunction expectation over position coordinates. The combination of wavefunction and ensemble average will be denoted by $\langle\langle \cdot \rangle\rangle $.”

11. Expanded discussion at start of section 4 to clarify the connection of the open systems theory with violations of the Kohn theorem.

12. Changed "expectation of momentum" to "expectation value of momentum" throughout.

13. In section 2.1, changed the wording to "The Gross-Pitaevskii equation may be generated by taking the functional derivative of the Gross-Pitaevskii Hamiltonian"

14. Added discussion at the end of section 2.1 clarifying the different averages used in the paper.

15. Added a concrete example for center of mass coordinate R, in text after equation (5).

16. We added a discussion of the cutoff and choice of representation in section 2.2.3

17. Fixed spelling of "Analagous"

18. Short additional discussion of conservation laws added at the end of section 2.2.1

19. Fixed wording "exists" => "exist" just below equation 22.

20. Romanised subscripts appropriately throughout.

21. Definition of Fourier transform given in Eq. (23).

22. Expanded discussion in section 3.3 regarding Eq. (41), clarifying the distinction between ensemble average and wavefunction average.

---

## Round 3 · Author Response

We first address objection

_3) the Kohn theorem is violated and this shows the theory is not appropriate for real experiments_

There are a number of systems under current and future study for which the theory is applicable. For our purpose it forms a very useful test for the formalism. We have revised the manuscript to clarify these points.

At the start of section 4 we have made substantial revisions of the text to point out several systems of interest where Kohn’s theorem is not satisfied by the system, and hence the SPGPE reservoir theory provides a useful approximation. We have also emphasized that the SPGPE has been used to give a first-principles treatment of high temperature non-equilibrium experiments. In particular, spontaneous vortex formation was described using one fitted parameter (ref [3] of revised manuscript), and the SPGPE gave a quantitative description of the experiment in ref [17] of the revised manuscript, with no fitted parameters:

"While the time-independent reservoir approximation (TIRA) is not strictly applicable for scalar BEC in the purely harmonic trap, there are a number of physical systems where it is applicable: Kohn’s theorem does not apply to a scalar Bose gas held in a harmonic trap if the trap becomes non harmonic at high energy. The theorem is also inapplicable for a harmonically trapped system if the reservoir consists of second atomic species confined by a different trapping potential, as may occur during sympathetic cooling. Furthermore, any system that is not harmonically trapped will not obey Kohn’s theorem, and is thus potentially amenable to the TIRA. Example systems in non-harmonic traps for which the theory is applicable include vortex decay in hard-wall confinement [46], soliton decay in a 1D toroidal trap (where the present approach was first used) [19], and persistent current formation in a 3D toroidal trap, where SPGPE simulations [17] compare well with experiment [44].

In this work our approach is simply to test the formalism on a simple model system within the TIRA by integrating out the spatial degrees of freedom to find effective stochastic equations of motion for the centre of mass. We stress that the TIRA approached used here is physically valid for (at least) two scenarios if immediate interest: non-harmonic trapping at high energies for a scalar BEC, and sympathetic cooling involving two BEC components in different harmonic traps [14]."

These points of context are also mentioned briefly in footnote 3 (page 14) of the revised manuscript.

We have also substantially revised our conclusions to further clarify the physical applicability of the theory of the harmonically trapped system to real physical systems. In particular:

“We tested our stochastic Ehrenfest equations in two ways. Considering the centre of mass motion of a finite-temperature quasi-1D condensate near equilibrium, we tracked the size of the largest projector corrections and saw they are indeed small. We also compared the steady-state correlations of position and momentum to analytic solutions derived by neglecting the projector corrections, finding excellent agreement. Our chosen test system has the weakness that the centre of mass motion in a purely harmonic trap is not strictly amenable to the reservoir theory due to a violation of Kohn's theorem. However, our treatment is physically relevant for non-harmonic trapping, multicomponent systems, and other systems of interest that physically violate Kohn's theorem, provided a low-energy fraction is harmonically trapped. Indeed, since the thermal equilibrium properties involve small excursions from equilibrium, the confining potential is only required to be _locally_ harmonic near the trap minimum and any number of non-harmonic effects may intrude at larger distances. The centre of mass motion thus provides an excellent formal and numerical test of the SERs, being one of the simplest states of motion to handle analytically. \par We have shown that SERs can be used to obtain analytic equations that agree with numerical solutions of the full SPGPE and offer some physical insight into the open system dynamics. Future work will explore systems involving analytically tractable excitations such as vortex decay in hard-wall confinement [46], soliton [39] and phase-slip dynamics [40] in toroidal confinement, sympathetic cooling [41,42], spinor BECs [23], and quantum turbulence in non-harmonic confinement [44-47].”

We hope that these changes are appropriate to address the remaining concerns regarding physical applicability.

_2) the equilibrium is not properly discusses and if it is the classical (field) equilibrium that I think is the case here, then it is actually not appropriate for the normal state of the gas, which the authors are discussing_

We find this comment unclear. We do not discuss the normal state of the gas directly, as our theory is a stochastic field theory of the low-energy partially degenerate fraction of the gas. There is a classical field equilibrium (within the truncated Wigner classical field approximation), that contains the condensate and a low energy normal fraction, but the latter must be extracted numerically (typically via Penrose-Onsager).

A high-energy normal fraction of the gas appears explicitly via the reservoir, due to the choice of the energy cutoff - it must be chosen to separate the coherent region from the incoherent region of phase space.

The normal fraction of the gas is thus included in the properties of the reservoir (gaussian statistics, chemical potential, temperature, and reservoir interaction rates), formulated using the single-particle Wigner function for the high energy part of the field.

In short, we are not sure we understand the question, but we would be happy to provide an answer to a clarified question.

With Best Regards,

Ashton Bradley (on behalf of the authors).

---

## Round 3 · List of Changes

Revisions:

  • Introduced acronym Stochastic Ehrenfest relation (SER)

  • Minor change of wording in second to last paragraph of introduction:

“As a test we apply the Ehrenfest relations to the centre of mass fluctuations of a harmonically trapped system tightly confined along two spatial dimensions. We find that the analtyic solution of the SER for the centre of mass is in close agreement with SPGPE simulations. ”

  • Replaced center —> centre throughout

  • Reordered the wording in the “v) Thermal equilibrium” paragraph at the end of section 3 on page 13, to improve clarity

  • Introduced acronym TIRA (Time Independent Reservoir Approximation)

  • Start of section 4, revised text:

"While the time-independent reservoir approximation (TIRA) is not strictly applicable for scalar BEC in the purely harmonic trap, there are a number of physical systems where it is applicable: Kohn’s theorem does not apply to a scalar Bose gas held in a harmonic trap if the trap becomes non harmonic at high energy. The theorem is also inapplicable for a harmonically trapped system if the reservoir consists of second atomic species confined by a different trapping potential, as may occur during sympathetic cooling. Furthermore, any system that is not harmonically trapped will not obey Kohn’s theorem, and is thus potentially amenable to the TIRA. Example systems in non-harmonic traps for which the theory is applicable include vortex decay in hard-wall confinement [46], soliton decay in a 1D toroidal trap (where the present approach was first used) [19], and persistent current formation in a 3D toroidal trap, where SPGPE simulations [17] compare well with experiment [44].

In this work our approach is simply to test the formalism on a simple model system within the TIRA by integrating out the spatial degrees of freedom to find effective stochastic equations of motion for the centre of mass. We stress that the TIRA approached used here is physically valid for (at least) two scenarios if immediate interest: non-harmonic trapping at high energies for a scalar BEC, and sympathetic cooling involving two BEC components in different harmonic traps [14]."

These points of context are also mentioned briefly in footnote 3 (page 14) of the revised manuscript.

  • Conclusions revised:

“We tested our stochastic Ehrenfest equations in two ways. Considering the centre of mass motion of a finite-temperature quasi-1D condensate near equilibrium, we tracked the size of the largest projector corrections and saw they are indeed small. We also compared the steady-state correlations of position and momentum to analytic solutions derived by neglecting the projector corrections, finding excellent agreement. Our chosen test system has the weakness that the centre of mass motion in a purely harmonic trap is not strictly amenable to the reservoir theory due to a violation of Kohn's theorem. However, our treatment is physically relevant for non-harmonic trapping, multicomponent systems, and other systems of interest that physically violate Kohn's theorem, provided a low-energy fraction is harmonically trapped. Indeed, since the thermal equilibrium properties involve small excursions from equilibrium, the confining potential is only required to be locally harmonic near the trap minimum and any number of non-harmonic effects may intrude at larger distances. The centre of mass motion thus provides an excellent formal and numerical test of the SERs, being one of the simplest states of motion to handle analytically. \par We have shown that SERs can be used to obtain analytic equations that agree with numerical solutions of the full SPGPE and offer some physical insight into the open system dynamics. Future work will explore systems involving analytically tractable excitations such as vortex decay in hard-wall confinement [46], soliton [39] and phase-slip dynamics [40] in toroidal confinement, sympathetic cooling [41,42], spinor BECs [23], and quantum turbulence in non-harmonic confinement [44-47].”

---

## Editorial Decision

published